# Bivalent antibody pliers inhibit β-tryptase by an allosteric mechanism dependent on the IgG hinge

Henry R. Maun[1], Rajesh Vij[2], Benjamin T. Walters [3], Ashley Morando[3], Janet K. Jackman[4], Ping Wu[5], Alberto Estevez[5], Xiaocheng Chen[2], Yvonne Franke[6], Michael T. Lipari[1], Mark S. Dennis[2], Daniel Kirchhofer [1], Claudio Ciferri[5], Kelly M. Loyet[3], Tangsheng Yi[4], Charles Eigenbrot [5], Robert A. Lazarus [1✉] & James T. Koerber [2✉]

Human β-tryptase, a tetrameric trypsin-like serine protease, is an important mediator of allergic inflammatory responses in asthma. Antibodies generally inhibit proteases by blocking substrate access by binding to active sites or exosites or by allosteric modulation. The bivalency of IgG antibodies can increase potency via avidity, but has never been described as essential for activity. Here we report an inhibitory anti-tryptase IgG antibody with a bivalency-driven mechanism of action. Using biochemical and structural data, we determine that four Fabs simultaneously occupy four exosites on the β-tryptase tetramer, inducing allosteric changes at the small interface. In the presence of heparin, the monovalent Fab shows essentially no inhibition, whereas the bivalent IgG fully inhibits β-tryptase activity in a hinge-dependent manner. Our results suggest a model where the bivalent IgG acts akin to molecular pliers, pulling the tetramer apart into inactive β-tryptase monomers, and may provide an alternative strategy for antibody engineering.

---

[1] Department of Early Discovery Biochemistry, Genentech, Inc., 1 DNA Way, South San Francisco, CA 94080, USA. [2] Department of Antibody Engineering, Genentech, Inc., 1 DNA Way, South San Francisco, CA 94080, USA. [3] Department of Biochemical and Cellular Pharmacology, Genentech, Inc., 1 DNA Way, South San Francisco, CA 94080, USA. [4] Department of Immunology Discovery, Genentech, Inc., 1 DNA Way, South San Francisco, CA 94080, USA. [5] Department of Structural Biology, Genentech, Inc., 1 DNA Way, South San Francisco, CA 94080, USA. [6] Department of Biomolecular Resources, Genentech, Inc., 1 DNA Way, South San Francisco, CA 94080, USA. ✉email: laz@gene.com; koerberj@gene.com

Mast cells play an essential role in the innate and adaptive immune responses to foreign stimuli such as allergens, bacteria, viruses, and venoms[1–3]. In addition to their protective role, mast cells have also been linked to a number of diseases including asthma, anaphylaxis, Crohn's disease, arthritis, and atherosclerosis[1,4]. Activation of mast cells induces the release of secretory granules that contain histamine, cytokines, chemokines as well as high levels of various proteases[3,5], in particular the trypsin-like serine protease tryptase, which is present in very high concentrations[6]. This burst of protease activity leads to many consequences such as increased collagen production by lung fibroblasts and proliferation and contraction of lung smooth muscle cells[4,7–9]. Since the majority of proteolytic activity stems from β-tryptases (highly similar βI-, βII- and βIII-isoforms), potent and specific inhibitors may have broad therapeutic potential for diseases driven by excessive mast cell activation[5].

Humans possess four major types of tryptases: the secreted, mast cell-expressed α- and β-tryptases, δ-tryptase, and membrane anchored γ-tryptase. Upon pro-peptide cleavage by cathepsin C, β-tryptase monomers form active tetramers in mast cell granules, where they are further stabilized by binding to serglycin proteoglycan carrying heparin glycosaminoglycan[3,10]. Since β-tryptase monomers are essentially inactive at physiological conditions, tetrameric β-tryptases are the predominant enzymatically active protease in mast cell granules[10–12]. A recent study has also shown that human α,β-heterotetrameric tryptases are also active proteases[13]. The activated tetramer stimulates a range of inflammatory and tissue remodeling processes through cleavage of a variety of substrates including proteinase-activated receptor (PAR-2), vasoactive intestinal peptide (VIP), matrix metalloproteinases (MMPs), fibronectin, and collagen[7–9,14]. The crystal structure of tetrameric β-tryptase shows a toroidal, donut-like homo-tetramer comprises monomers that contact their neighbors at both the large and small interfaces (Supplementary Fig. 1)[15,16]. Tetramers are further stabilized by heparin, which is negatively charged and thought to bind to a large positively charged surface that spans both of the small interfaces[17,18]. While the high heparin concentration in mast cell granules likely aids in tetramer formation, the low extracellular heparin concentration may serve as a natural inactivation mechanism, explaining the relatively short serum half-life of ~2 h for active β-tryptase tetramers[19]. A detailed mechanistic assessment of the role of heparin in tetramer stability, monomer/tetramer equilibria and enzymatic activity has been previously described[12]. An excellent review on tryptases provides additional insights into these aspects[20].

The β-tryptase tetramer contains four active sites, located inside the core of the tetramer and facing the donut hole. Furthermore, the small pore entrance (15 Å × 40 Å) limits substrates to small peptides and unstructured loops[15,16]. To date, no endogenous mammalian serine protease inhibitors (e.g., serpins like α1-antitrypsin, α2-macroglobulin or Kunitz- or Kazal-domain containing proteins) are known to inhibit the proteolytic activity of β-tryptase. The only known natural macromolecular inhibitors of β-tryptases are leech-derived tryptase inhibitor (LDTI), tick-derived protease inhibitor (TdPI), and lactoferrin[21–23]. Both LDTI (4.7 kDa) and TdPI (11.1 kDa) have the ability to block two or three of the four active sites of the β-tryptase tetramer, respectively. Engineered cysteine knot miniproteins derived from MCoTI-II have also been reported as potent β-tryptase inhibitors[24]. However, none of these molecules are selective inhibitors of β-tryptase. Efforts to develop small molecule β-tryptase inhibitors have shown therapeutic effects in guinea pig and sheep asthma models and in humans[25]. However, poor potency, selectivity, oral bioavailability and/or tolerability of these small molecules have created challenges, leading to toxicity and program termination. We therefore sought an antibody as a selective inhibitor of β-tryptase with drug-like properties.

Antibodies possess exquisite specificity for their target and can be potent inhibitors of protease activity. Existing antibodies that inhibit protease activity fall into two broad classes: direct active site blockers or indirect modulators. Notably, many of the direct active site blockers (e.g., anti-HGFA, -uPA, and -matriptase) do so by inserting a long complementarity-determining region (CDR) H3 in the vicinity of the active site[26–28]. Indirect antibodies (e.g., anti-HGFA, -hepsin, -BACE1, and -MMP9) employ various mechanisms ranging from allosteric effects on the active site, inhibiting binding at an exosite, or preventing zymogen activation[29–31]. Given the structural constraints on active site access for β-tryptase, we hypothesized that anti-tryptase inhibitory mAbs would most likely use an indirect mechanism, although a very long CDR could potentially reach into the pore to directly contact the active site. In fact, we have recently described a mouse-derived anti-tryptase antibody in which Fab binding allosterically modulates the large and small interfaces of the β-tryptase tetramers to dissociate the tetramer into inactive monomers[32]. While this antibody has promising activity both in vitro and in vivo, antibodies with different epitopes, binding modes or mechanisms of action (MOAs) are highly desirable. To explore this strategy, we employed rabbit immunizations for additional antibody diversity as rabbits have been shown to generate antibodies with longer CDRH3s and CDRL3s and thus might be able to inhibit β-tryptases using other mechanisms[33].

Here, we describe the MOA of an inhibitory anti-tryptase antibody (E104) through a series of biochemical and structural biology experiments. We demonstrate that this antibody potently inhibits β-tryptase in both biochemical and cell-based assays. We employ electron microscopy (EM), X-ray crystallography and hydrogen deuterium exchange mass spectrometry (HDX or HX MS) to map the β-tryptase:Fab interactions and elucidate an allosteric mechanism of inhibition. Notably, the monovalent Fab is unable to inhibit the enzymatic activity of tetrameric tryptase due to the stabilizing effect of heparin, whereas the bivalent E104 IgG potently dissociates tetrameric β-tryptase and inhibits enzymatic activity. We further establish the molecular features of the IgG required for this inhibitory activity. Finally, we propose that the predominant mode of inhibition occurs through the bivalent IgG working as molecular pliers that simultaneously bind and pry apart two protomers within the tetramer to dissociate the tetramer into monomers and inactivate the enzyme.

## Results

**Identification of an inhibitory anti-tryptase mAb.** Given that the active site of β-tryptase is not readily accessible to macromolecules and that rabbit antibodies possess long CDRs that might enable additional MOAs, we generated a panel of anti-tryptase antibodies via immunization of rabbits followed by standard hybridoma technology[32]. Purified clones were subsequently characterized for binding to monomeric β-tryptase by ELISA and for inhibitory activity against the active tetrameric β-tryptase using a chromogenic enzyme assay in the presence of heparin to mimic physiological conditions. One antibody (E104) exhibited potent inhibition of β-tryptase with an IC$_{50}$ of 4 nM; similar potencies were observed for rabbit Fab-human Fc chimeras (chIgG1 and chIgG4) (Fig. 1a).

We humanized E104 by grafting complementarity-determining regions (CDRs) into the consensus human VL$_{KI}$ and VH$_{IV}$ acceptor frameworks and evaluating key Vernier/framework residues required for binding as described in materials and methods (Supplementary Fig. 2). We expressed and purified human IgG1 forms for twelve variants and measured their

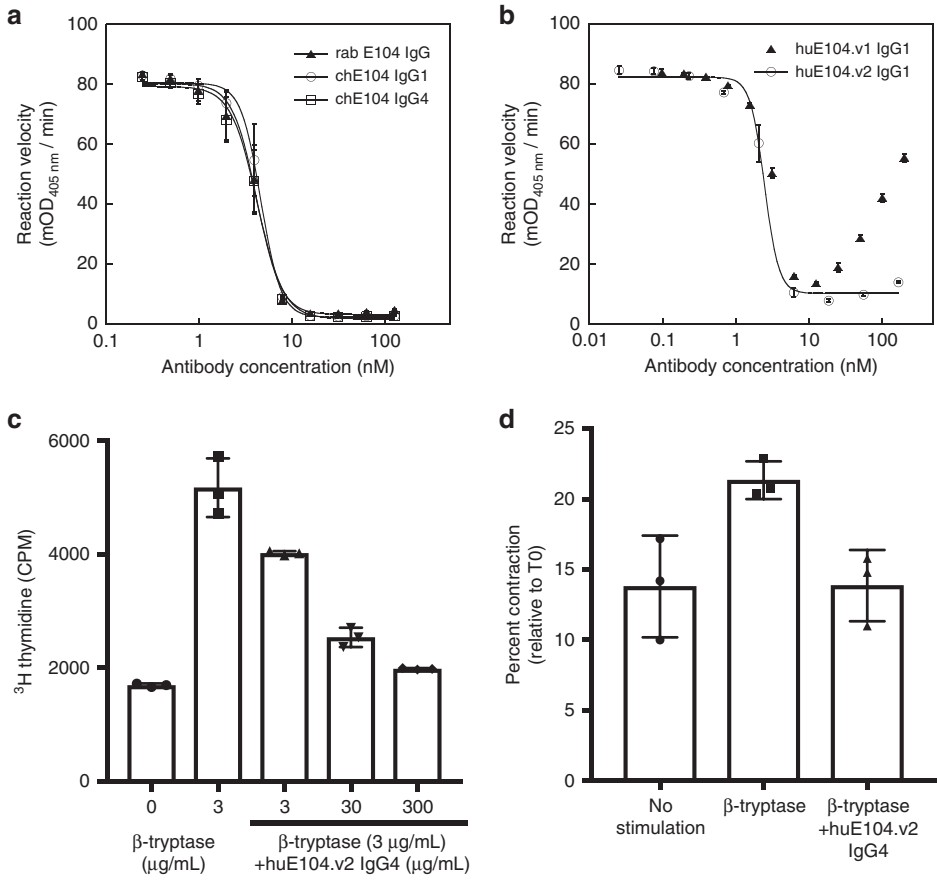

**Fig. 1 Anti-tryptase antibody E104 inhibits β-tryptase in biochemical and cell-based assays. a** Concentration-dependent inhibition of βI-tryptase enzymatic activity in the presence of heparin by rabbit (rab) E104, rabbit-human chimeric (ch) E104 IgG1 and E104 IgG4. **b** Humanized E104.v1 and E104.v2 IgG1 show concentration-dependent inhibition of βI-tryptase enzymatic activity, but E104.v1 loses its inhibitory activity at higher concentrations (hook effect). **c** Primary human bronchial smooth muscle cell (BSMC) proliferation stimulated by βI-tryptase, which is inhibited by anti-tryptase humanized E104.v2 IgG4 in a concentration-dependent manner. **d** Primary human smooth muscle cell contraction is induced by βI-tryptase, which is inhibited by humanized E104.v2 IgG4. Data is represented as mean ± s.d. ($n = 3$ biologically independent experiments; **a**, **b** and $n = 4$ biologically independent experiments; **c**, **d**). Graphs were created using Kaleidagraph v4.1.3. Source data are provided as a Source Data file.

### Table 1 Binding affinities of E104 IgG variants.

| Name | Sequence | $K_D$ (nM) |
|------|----------|-----------|
| IgG1.v1 | EPKSCDKTH---------T | 0.5 ± 0.02 |
| IgG1.v2 | EPKSCDKTH---------T | 0.6 ± 0.01 |
| IgG2 | ERKCCV-----------E | 0.6 ± 0.1 |
| IgG4 | ESKYGP-----------P | 0.7 ± 0.04 |
| G1 + G | EPKSCDKTHG--------T | 0.5 ± 0.01 |
| G1 + GG | EPKSCDKTHGG-------T | 0.4 ± 0.02 |
| G1 + GGS | EPKSCDKTHGGS------T | 0.5 ± 0.05 |
| G1 + (GGS)₃ | EPKSCDKTHGGSGGSGGST | 0.5 ± 0.1 |
| G1-H | EPKSCDKT----------T | 0.6 ± 0.1 |
| G1-TH | EPKSCDK-----------T | 0.5 ± 0.07 |
| G1-KTH | EPKSCD------------T | 0.5 ± 0.09 |
| G1.G5 | EPKSCGGGG---------G | 0.6 ± 0.1 |
| G1.P5 | EPKSCPPPP---------P | 0.4 ± 0.1 |
| G1.C | EPKSCDKCH---------T | 0.3 ± 0.01 |
| G1.CC | EPKSCDKCC---------T | 0.4 ± 0.03 |

Upper hinge sequences of E104.v2 IgG variants and binding affinities to βI-tryptase monomer. Sensorgrams from six representative antibodies are shown in Supplementary Fig. 3. Affinities were measured by surface plasmon resonance at 25 °C ($n = 2$). Source data are provided as a Source Data file.

binding affinity to βI-tryptase monomer by surface plasmon resonance to identify the two highest affinity variants (E104.v1 and E104.v2), which have identical CDRs, differing only at two framework positions in the heavy chain (V71R and F78V, respectively). We first confirmed that these two variants had similar binding affinities ($K_D$) as measured by surface plasmon resonance (Table 1 and Supplementary Fig. 3) and similar inhibitory properties by determining their $IC_{50}$ values (Fig. 1b). Both mAbs bound with a $K_D$ of ~0.5 nM to βI-tryptase monomer and exhibited $IC_{50}$ values of 2.5 nM. E104.v1 exhibited a pronounced bell-shaped curve or hook effect in which the inhibitory effect decreased at supersaturating antibody concentrations, indicating that the two framework residue differences between v1 and v2 alter the inhibitory activity of the IgG.

We next examined whether tryptase-dependent cellular responses such as proliferation or cell-collagen matrix contraction could be inhibited by E104.v2 IgG. Human bronchial smooth muscle cells (BSMC) were grown in culture plates without serum for 24 h prior to treatment with 100 nM human βI-tryptase and different concentrations of E104.v2 IgG4 (3–300 μg/mL). Proliferation was measured by monitoring incorporation of ³H-labeled thymidine for 6 h following incubation with β-tryptase and antibody. E104.v2 exhibited a dose-dependent inhibitory effect and a dose of 300 μg/mL fully inhibited the cellular proliferation induced by active β-tryptase (Fig. 1c). We next grew

**Table 2 Summary of data collection and structural refinement.**

|  | E104.v1 Fab:βI-tryptase |
|---|---|
| PDB code | 6VVU |
| Data collection |  |
| Space group | P2₁ |
| Cell dimensions |  |
| $a, b, c$ (Å) | 89.57, 168.81, 114.65 |
| $α, β, γ$ (°) | 90, 109.97, 90 |
| Resolution (Å) | 50–3.0 (3.112–3.005) |
| $R_{sym}$ or $R_{merge}$ | 0.098 (0.577) |
| $I/σI$ | 11.3 (2.1) |
| Completeness (%) | 97.8 (99.7) |
| Redundancy | 3.4 (3.5) |
| Refinement |  |
| Resolution (Å) | 50–3.0 (3.112–3.005) |
| No. of reflections | 62353 (1266) |
| $R_{work}/R_{free}$ | 0.187/0.232 |
| No. of atoms |  |
| Protein | 20,689 |
| Ligand/ion | 3 |
| Water | 2 |
| B-factors |  |
| Protein | 71.6 |
| Ligand/ion | 48.4 |
| Water | 71.6 |
| R.m.s. deviations |  |
| Bond lengths (Å) | 0.01 |
| Bond angles (°) | 1.2 |

Values in parentheses are for highest-resolution shell.

human BSMC in a three-dimensional (3D) collagen matrix and measured the changes in contraction induced by active β-tryptase in the presence and absence of antibody. β-tryptase increased the level of collagen contraction by ~50% and addition of E104.v2 IgG4 was sufficient to completely block this effect (Fig. 1d). In summary, the E104 antibody inhibits β-tryptase in both biochemical and cell-based assays.

**Structural analysis of E104 and βI-tryptase complex.** We reasoned that structural characterization of the E104:β-tryptase complex would help inform the potential MOA. Owing to challenges associated with crystallizing full-length IgGs, we employed the E104 Fab for all structure studies. To determine the location of the Fabs on the tetramer, we isolated a stable complex of βI-tryptase with E104.v1 Fab, which does not dissociate the tetramer (vide infra), by size-exclusion chromatography (SEC) and analyzed the complex by electron microscopy (EM). A representative EM micrograph showing individual particles is shown in Supplementary Fig. 4 and data collection, refinement and validation statistics are shown in Supplementary Table 1. We obtained an EM reconstruction by negative staining at ~15 Å, which showed four copies of E104.v1 Fab where each Fab interacts essentially with only one of the four β-tryptase protomers. Since the β-tryptase protomers associate into a ring-like tetramer in an up-down-up-down configuration, there are two Fabs projecting up (bound to protomers A and C that are diagonally across the tetramer) and two Fabs projecting down (bound to protomers B and D) from the tetramer (Supplementary Fig. 5).

To gain further insights into the molecular details of this complex, we solved the crystal structure of the wild-type (WT) tetrameric βI-tryptase bound to E104.v1 Fab at 3.0 Å resolution (Table 2). A portion of the electron density map including the contour level is shown in Supplementary Fig. 6. Despite extensive

efforts, we were unable to crystallize E104.v2 with βI-tryptase. The crystallographic asymmetric unit contains one βI-tryptase tetramer interacting with four copies of E104.v1 Fab (Fig. 2a, b) in accord with the EM structure. Overall, each of the four copies of the β-tryptase protomer and its partner Fab are very similar, having main chain rmsds from 0.85–1.44 Å and slight variations in Fab elbow angles (138–152°) (Supplementary Fig. 7). Overall, the antibody-antigen interface is characterized by contributions primarily from the heavy chain (~71% of buried surface area) and is on the small side for antibody-antigen complexes (682 Å² vs. ~1100 Å² on average)[34]. On the antibody side, key interactions are made with CDRL1 (Y29, N30, R32), CDRL3 (R94), CDRH1 (Y32), CDRH2 (S52-A54, T56, F58) and CDRH3 (P96-Y99, R100e) (Fig. 2c). On the β-tryptase side, key interactions are made with W38, Q50, D60e, L61, A62, R65, Q81, L82, L83, P84, V85, S86, R87, E107, L108, E109 and E110 (Fig. 2d). This epitope lies close to the small interface (protomers A:B or C:D). Upon binding of E104.v1 Fab to βI-tryptase, we observed very similar overall conformations as found with βII-tryptase and βI-tryptase bound to another inhibitory mAb (31A.v11) (rmsds of 0.47 Å and 1.0 Å, respectively) (Supplementary Fig. 8)[15,32]. Furthermore, comparison to the βII-tryptase structure that lacks a bound antibody revealed no conformational changes in tryptase upon E104.v1 Fab binding that would explain the inhibitory effect. The E104 epitope largely overlaps with that of the 31A.v11 Fab epitope, except that the epitope for E104.v1 is shifted away from V60c-A63 and more towards Q81-R87 (Fig. 2e). Overall, this positions the Fab closer towards the small interface of the tetramer. In summary, while our crystal structure revealed a 4:4 Fab:protomer stoichiometric architecture, it did not elucidate the antibody MOA as we had hoped.

As an alternative strategy to characterize the E104:β-tryptase interaction, we performed HDX-MS experiments with monomeric βI-tryptase alone and bound to either E104.v1 or E104.v2 Fab. We focused on both Fabs since our biochemical inhibition data suggested that v1 IgG loses activity at high concentrations where monovalent Fab binding predominates. Amide protons with slower exchange rates in the presence of antibody indicate that these regions are buried upon Fab binding or allosterically altered compared to β-tryptase alone. Not surprisingly, sets of the β-tryptase amide protons that showed slower exchange when bound to E104.v1 or E104.v2 were similar (orange color in Fig. 2f) and generally agreed with contact residues identified in the crystal structure (Supplementary Table 2). However, we also observed a subset of amide protons that showed slower exchange that were unique to either E104.v1 or E104.v2 bound forms. The largest unique differential effect was found on a loop that contains two critical residues (Y74 and Y75) involved in stabilization of the small interface, whereby substantially greater retardation in the exchange rate was observed when bound to E104.v2 (Supplementary Note 1). These results suggest that changes in two framework residues (V71R and F78V) generate a distinct effect on β-tryptase. Furthermore, since the interface is symmetric, residues from both interacting subunits are affected when E104.v2 is bound to each subunit, which would amplify any instability at the small interface.

**Biochemical analysis of E104 Fab and βI-tryptase complexes.** To explore the relevance of these allosteric changes on Fab function, we analyzed mixtures of the E104 Fabs and tetrameric βI-tryptase in the absence of heparin by SEC as previously described[11]. Tetrameric βI-tryptase with a molecular weight of 120 kDa had an elution volume ($V_e$) of 13 mL (peak two; black curve in Fig. 3a), which is consistent with previous data[11]. The mixture of tetrameric βI-tryptase and E104.v1 Fab (orange curve)

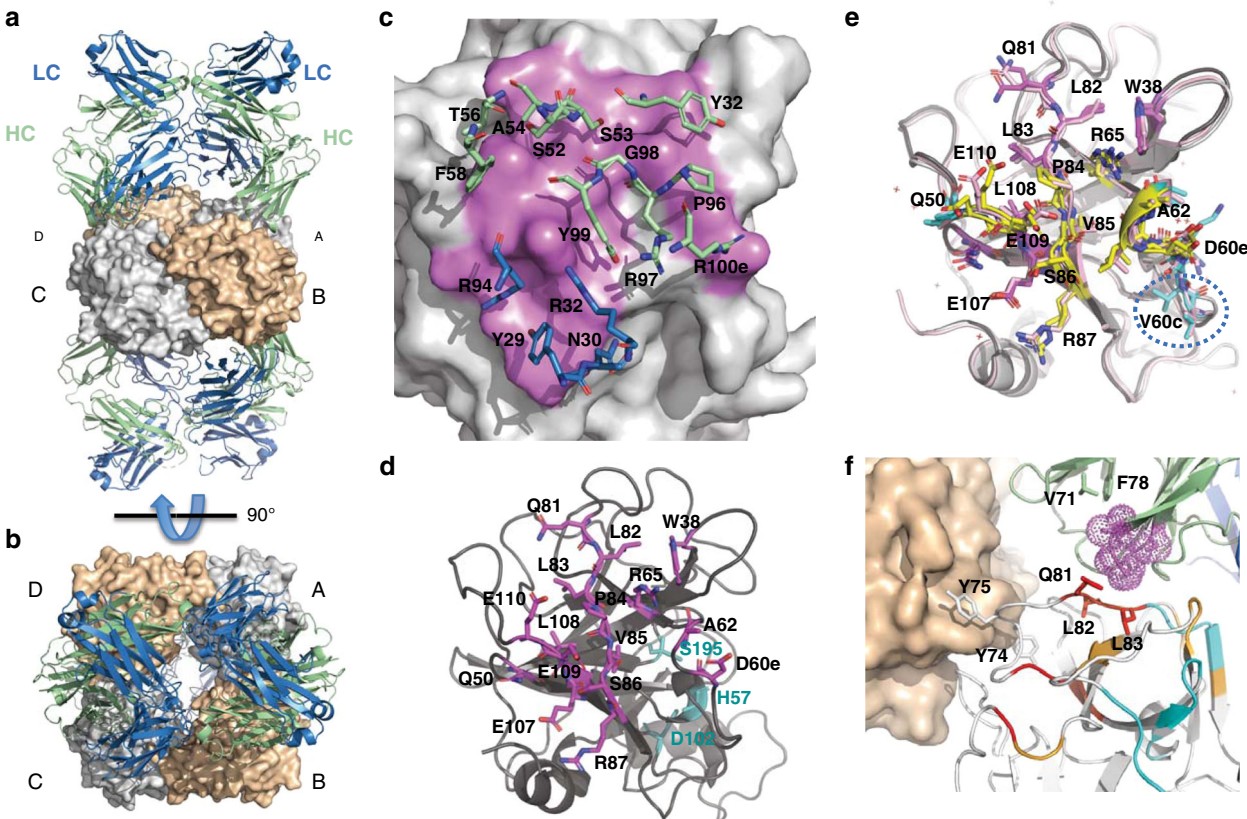

**Fig. 2 X-ray structures of tetrameric βI-tryptase in complex with E104.v1 Fab. a** Four Fabs simultaneously bind to tetrameric βI-tryptase, one Fab per protomer. Top Fabs bind to protomers A and C (white) and bottom Fabs bind to protomers B and D (beige); heavy (HC) and light chain (LC) are colored green and blue, respectively. **b** Top-down view of **a** showing Fabs binding to protomers A and C. **c** The paratope residues of the HC and LC within 4 Å of βI-tryptase (white surface) are shown as sticks and colored in green and blue, respectively. The epitope on βI-tryptase is colored violet. **d** Residues in βI-tryptase monomer within 4 Å of the Fab are shown as sticks and colored violet. Catalytic triad residues (H57, D102, and S195) are shown as sticks and colored teal. **e** Structurally aligned βI-tryptase monomers (cartoon) from βII-tryptase (pink), βI-tryptase bound to 31A.v11 Fab (dark gray), and βI-tryptase bound to E104.v1 Fab (light gray) exhibit similar conformations. Epitope residues that are unique to 31A.v11 (cyan), unique to E104.v1 (magenta), or shared (yellow) are colored accordingly. Only 31A.v11 induces a conformational change in V60c (dashed blue circle) that leads to destabilization of the large interface. **f** βI-tryptase peptide bonds with protons in slower H-D exchange due to E104.v1 Fab binding are colored orange (affected by both v1 and v2), cyan (affected by v1 only) and red (affected by v2 only). Tryptase protomers A (or C) and B (or D) are shown in white and beige, respectively. Vernier residues V71 and F78 are shown as sticks and CDRH2 residues are shown as violet spheres. Tryptase residues Q81-L83 that are within 4 Å of CDRH2 are shown as sticks. Tryptase key contact residues Y74 and Y75 in the small interface (protomers A-B or C-D) are shown as sticks. Individual structural images were created using PyMOL v2.2 and panels combined using Adobe Illustrator v16).

showed a protein peak with a smaller $V_e$ of 10.8 mL (peak one in Fig. 3a) containing both Fab and β-tryptase as determined by sodium dodecyl sulphate–polyacrylamide gel electrophoresis (SDS-PAGE) (Fig. 3d). This result suggests a stable complex between the Fab and β-tryptase tetramer in a 4:1 ratio, respectively, as observed in our crystal structure. We conclude that E104.v1 Fab does not affect tetramer stability. In stark contrast to the v1 Fab, the mixture of tetrameric βI-tryptase and E104.v2 Fab showed a complex on SEC (peak 5, $V_e = 13$ mL, orange curve in Fig. 3b), which migrated similarly to a complex of monomeric β-tryptase zymogen and E104.v2 Fab (peak 4, $V_e = 13.1$ mL, blue curve in Fig. 3b) as confirmed by SDS-PAGE (Fig. 3e). We conclude, that in the absence of heparin, E104.v2 Fab binding induces dissociation of the tetramer into monomers. Stable tryptase dimers were not observed. This difference between E104.v1 and E104.v2 Fabs, which is likely driven by the allosteric changes we observed by HDX-MS, nicely explains lack of inhibition observed for the E104.v1 IgG (Fig. 1b) at high concentrations of IgG, where monovalent Fab binding to β-tryptase predominates. It also explains why we were unable to obtain a structure of tetramer with E104.v2. However, since both v1 and

v2 IgGs inhibit tryptase up to ~20 nM IgG (Fig. 1b), these results indicate that some other primary mode of inhibition occurs.

To test whether E104.v2 Fab binding only affects small tetramer interfaces, we analyzed Fab:tryptase complexes using βI-tryptase tetramers with engineered disulfide bonds in the small interface (Y75C) or large interface (I99C)[11]. The E104.v2 Fab:β-tryptase complex generated with Y75C migrated similarly to E104.v1 Fab:tetrameric β-tryptase (peak 7, $V_e = 10.8$ mL; brown curve in Fig. 3c), whereas the I99C mutant:E104.v2 Fab complex migrated between a tetramer and a 1:1 Fab:monomer complex, suggesting a 2:1 Fab:dimeric β-tryptase complex (peak 8, $V_e = 11.9$ mL; blue curve in Fig. 3c) as confirmed by SDS-PAGE (Fig. 3f, g). These data indicate that under these conditions E104.v2 Fab destabilizes the small interface, which leads to tetramer dissociation. This finding agrees well with our structural data (Fig. 2) demonstrating that the epitope is directly adjacent to the small interface.

**Effects of heparin on inhibitory activity of Fab and IgG.** Our SEC data suggests that locking the small interface via an engineered disulfide prevented the E104.v2 Fab from dissociating the

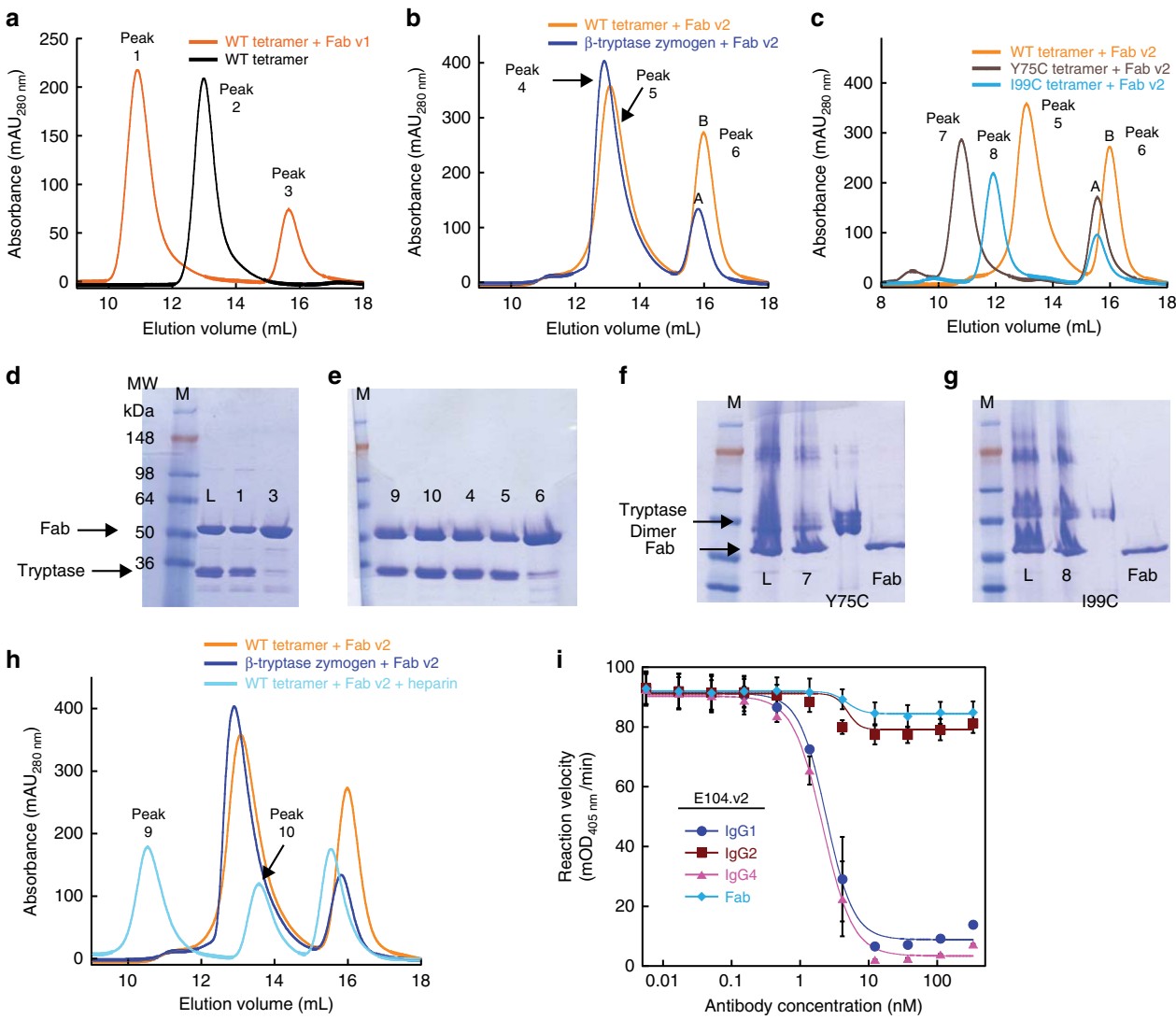

**Fig. 3 Complex formation and SEC of βI-tryptase WT and mutants Y75C and I99C in complex with Fabs E104.v1 or E104.v2 in the absence and presence of heparin (100 μg/mL). a** SEC of βI-tryptase tetramer alone (black) and in complex with excess E104.v1 Fab (orange). **b** SEC of βI-tryptase tetramer in complex with E104.v2 Fab without heparin (orange) and zymogen monomeric βI-tryptase in complex with Fab E104.v2 (blue). **c** SEC of E104.v2 Fab in complex with WT (orange) and mutant βI-tryptase tetramers Y75C (brown) and I99C (light blue). Elution volumes of the marked peaks 1–8 were: Peak 1 = 10.8 mL, Peak 2 = 13 mL, Peak 3 = 15.8 mL, Peak 4 = 13 mL, Peak 5 = 13.1 mL, Peak 6 = 15.8 mL, Peak 7 = 10.8 mL, Peak 8 = 11.9 mL, Peak 9 = 10.5 mL, Peak 10 = 13.6 mL. Peaks 6A and 6B have elution volumes of 15.6 and 15.8 mL, respectively, and represent excess Fab. **d–g** SDS-PAGE analysis from the middle of the peaks 1 through 10. M, L and numbers 1 through 10 refer to Markers, Load and Peaks 1 through 10. **h** SEC of βI-tryptase tetramer in complex with Fab E104.v2 with (cyan) and without heparin (orange) and zymogen monomeric βI-tryptase in complex with Fab E104.v2 (blue). Gels show representative results from two independent experiments. **i** Concentration-dependent inhibition of βI-tryptase enzymatic activity in the presence of heparin by E104.v2 as IgG1, IgG2, IgG4 and Fab. Data is represented as mean ± s.d. (n = 3 biologically independent experiments). Graphs were created using Kaleidagraph v4.1.3. Source data are provided as a Source Data file.

tetramer (Fig. 3c). Under physiological conditions, heparin plays an analogous role where it stabilizes the tetrameric state of active β-tryptase via interactions with a positive patch near the small interfaces[17,18]. To determine if this interaction inhibits dissociation with the E104.v2 Fab, we further analyzed the E104.v2:tetrameric tryptase complex in the presence of heparin by SEC. Interestingly, we observed two peaks, one corresponding to Fab: monomeric tryptase (peak 10, $V_e$ = 13.6 mL; cyan curve in Fig. 3h) and a second corresponding to a stable Fab:tetrameric tryptase complex (peak 9, $V_e$ = 10.5 mL). We conclude that heparin partially neutralizes the dissociating effect of E104.v2 Fab. While both heparin and the Fab bind near the small interface, these epitopes do not overlap. Based on these findings, we

postulated that heparin would neutralize the Fab inhibitory activity in our enzyme assay.

To evaluate this hypothesis, we assayed the E104.v2 Fab alongside the IgG1, IgG2, and IgG4 isotype versions. We included IgG2 and IgG4 isotypes since these have been historically employed for therapeutic applications in which only target neutralization is required (e.g., blocking a cytokine). As predicted from our SEC data, the Fab failed to inhibit β-tryptase in enzymatic assays that contain heparin (Fig. 3i). While the IgG1 and IgG4 versions of E104.v2 potently inhibited β-tryptase activity in the presence of heparin, it was surprising that the IgG2 version did not, despite possessing a similar affinity for monomeric β-tryptase as the inhibitory IgG1 and IgG4 versions

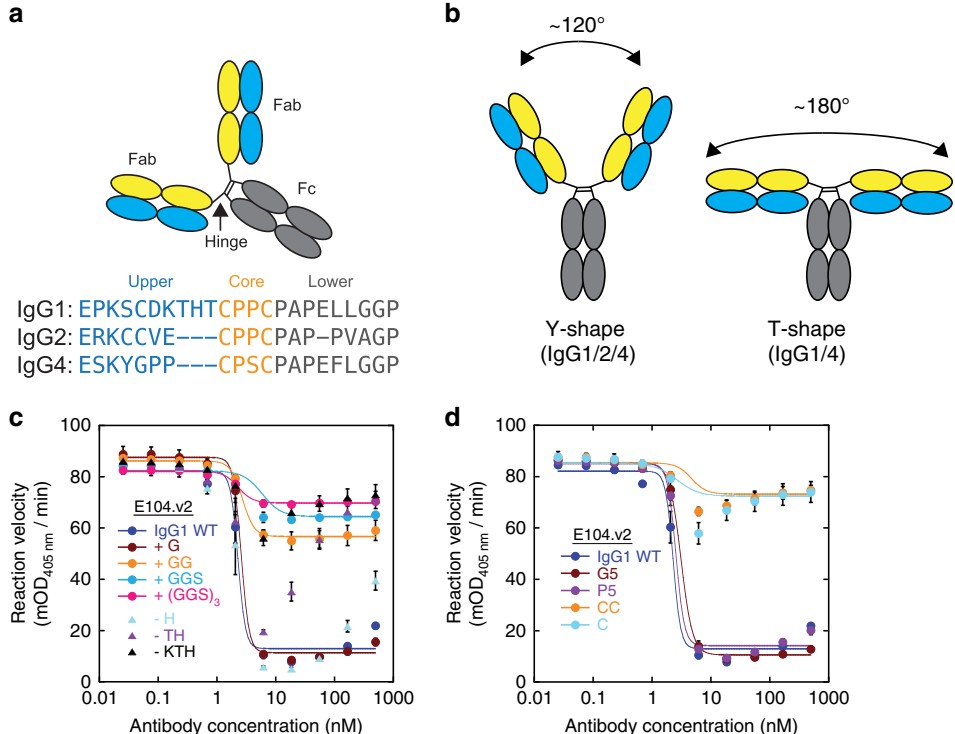

**Fig. 4 IgG hinge architecture controls E104 inhibitory effect on β-tryptase. a** Cartoon representation of IgG and sequence comparison of the hinge regions of human IgG1, 2, and 4. **b** Cartoon depiction of Fab wingspans for IgG1, 2, and 4. IgG1 and 4 exhibit wingspans between 120° and 180° (Y to T shape) while IgG2 has a narrower wingspan around 120° (predominantly Y shape). **c** Inhibition of β-tryptase activity by E104.v2 IgG1 having extended (+G, +GG, +GGS or +(GGS)₃) and shortened (-H, -TH, -KTH) hinges. **d**, Inhibition of β-tryptase activity by E104.v2 IgG1 having different flexibilities (G5 vs. P5) or constraints (C, CC). Data is represented as mean ± s.d. ($n = 3$ biologically independent experiments). Graphs were created using Kaleidagraph v4.1.3 and images were created in Adobe Illustrator v16. Source data are provided as a Source Data file.

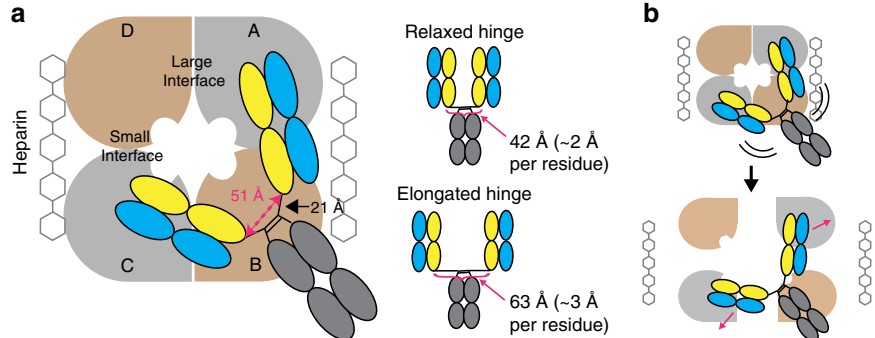

**Fig. 5 E104.v2 acts as molecular pliers to dissociate tetrameric β-tryptase. a** Cartoon depiction of E104.v2 IgG complex with tetrameric β-tryptase. The distance between Fab C-termini is 51 Å (magenta dashed line), as measured in our βI-tryptase:E104.v1 structure, and the distance spanned by upper hinge region to connect each Fab to the hinge disulfides and Fc is 21 Å (black line). Cartoon depictions of the IgG with two relaxed hinges, which separates two Fabs by 42 Å, and two fully elongated hinges, which separates two Fabs by 63 Å are shown. HC and LC are colored yellow and blue, respectively. The large and small interfaces are labeled and heparin bound across the small interface is depicted. **b** Cartoon depiction of E104.v2 IgG acting as reverse molecular pliers to dissociate the heparin-bound tetramer into monomers by binding to two diagonally opposed protomers; Fabs on top bind to protomers A and C (light gray), which are diagonally across from each other. Images were created in Adobe Illustrator (v16).

(Table 1). These results show that both bivalency and a particular IgG architecture are required for the inhibitory activity of E104.

**Impact of IgG architecture on E104 activity**. We next sought to identify the molecular features of the IgG architecture required for activity. Given that Fc-mediated effects are likely not required for inhibition, we focused our attention on the hinge region of the antibody. This region serves as the linker between the two Fab arms and the Fc and the upper hinge controls the flexibility and orientations of the Fab arms relative to each

other (Fig. 4a)[35,36]. Unlike the upper hinges of IgG1 or IgG4, the upper hinge of IgG2 has a unique cysteine that pairs with the corresponding cysteine in the other upper hinge. These differences constrain the motion of the two Fabs arms and result in more rigid and shorter Fab-Fab wingspans for the IgG2 isotype that lock it predominantly into a Y shape, whereas the flexibility in IgG1 and IgG4 isotypes enables them to adopt both a Y shape and a more extended T shape (Fig. 4b)[35]. To study both the role of hinge length and rigidity on the IgG activity, we designed a panel of E104.v2 variants with different

hinges (Table 1). Starting with the WT IgG1 hinge, we made three classes of IgG variants: (i) different hinge length, (ii) different rigidity (G5, P5), and (iii) constrained (C, CC). We generated the rigidity variants (G5 and P5) by replacing the hinge with five glycines or prolines, respectively. The constrained variants (C and CC) were generated by replacing one or two residues in the hinge with cysteines. As expected all variants exhibited similar affinities to βI-tryptase monomer by SPR (Table 1). While E104.v2 IgG1 tolerated a single glycine insertion into the linker region without any loss in inhibition, the addition of two glycines led to a partial extent of inhibition of 32% and even longer insertions abolished most of the inhibitory activity (Fig. 4c). Furthermore, shortening the hinge length by 3 residues (-KTH) also greatly reduced the extent of inhibition (black triangles in Fig. 4c). These combined results indicate that an upper hinge length of 8–12 residues (~17–26 Å, based on hinge length in full-length hIgG1, 1HZH[37]) is required for full antibody activity. Rigidity of the upper hinge did not have an impact since both fully flexible (G5) and more rigid (P5) hinges retained activities that are comparable to those of the WT IgG1 hinge (Fig. 4d). Finally, the introduction of cysteines into the IgG1 hinge to constrain it in a similar manner as IgG2 abolished nearly all of the inhibitory activity (Fig. 4d). Overall, these data indicate that E104.v2 requires an IgG format with an unconstrained upper hinge of ~17–26 Å to effectively inhibit β-tryptase activity.

**Proposed model of inhibition by E104.** To understand how the bivalent IgG architecture controls the activity, we constructed a model of full-length E104 IgG1 based on our E104.v1 Fab:βI-tryptase complex structure, in which two Fabs are bound to diagonally opposed tryptase protomers. The final resolved C-terminal HC residue in the E104.v1 Fab is P217 and P217 is separated by 51 Å from the P217 in the adjacent copy of E104. v1 Fab (magenta dashed line in Fig. 5a). However, the remaining portion of the hinge is not present in the Fab. To build this additional segment into our model, we used the upper hinge segment (P217-C226) from the existing full-length IgG1 (PBD 1HZH)[37]. This upper hinge segment spans 21 Å and thus the two Fab C-termini (i.e., P217-P217) could be maximally separated by ~42 Å (relaxed hinge in Fig. 5a). This linker length, in which each residue spans ~2 Å, is similar to that observed in natural relaxed linkers that connect domains within a protein[38]. In contrast, modeling indicates that fully elongated linkers in a tense conformation can span ~3 Å per residue[38] and thus both hinges could span up to 63 Å (42 Å x 3/ 2). This elongated hinge would create significant strain between the two Fab arms and could pry apart two protomers in the tetramer. As the hinge length increases, eventually both hinges can span the Fab C-termini distance in a more relaxed conformation, which would be predicted to reduce the ability of the IgG to dissociate the active tetramer. In support of this, we found that inserting at least three residues, such that both hinges can now span >54 Å in a relaxed state, greatly reduced the extent of inhibitory activity of E104 (Fig. 4c). Additionally, removing three residues from the hinge, such that both hinges span <51 Å in a fully extended form, also greatly reduced the extent of inhibitory activity. Our data supports a model where the IgG works akin to a pair of molecular pliers where both Fab arms, which are connected via a strained hinge, simultaneously bind to two protomers diagonally across the tetramer and pry them apart, resulting in tetramer dissociation (Fig. 5b). Since E104.v1 IgG, which exhibits no allosteric effect on the small interface, has a similar IC$_{50}$ as E104.v2 IgG, we hypothesize

that the molecular pliers MOA predominates under most conditions except at very high IgG:β-tryptase ratios.

## Discussion

The majority of the proteolytic activity released by mast cells in healthy and diseased states stems from β-tryptases. Since inflammatory diseases including asthma, anaphylaxis, and others may be due in part to aberrant mast cell activation, inhibitors of β-tryptases have broad therapeutic potential. However, other proteases in mast cells such as chymase and carboxypeptidase A3 also play significant roles in mast cell biology and perhaps disease as well[1,3,4]. Recent reports have shown that mast cell proteases (tryptase and/or chymase) can cleave a variety of cytokine and chemokine substrates, including several well-known TH2-promoting cytokines IL-33, IL-18, TSLP, IL-15, and IL-21[39,40]. How any downstream signaling by altered levels of these cytokines might affect any possible therapeutic benefit from anti-tryptase awaits further studies in patients.

The active tetrameric β-tryptase adopts a toroid structure with neighboring monomers arranged in an alternating up-down-up-down conformation with the active sites of the four protomers residing inside the pore. This architecture excludes potential macromolecular inhibitors from accessing the active site. Here, we characterize a highly selective and potent antibody that inhibits the enzymatic activity of human β-tryptase through tetramer dissociation via a mechanism in which a flexible, yet specific, bivalent IgG architecture is essential.

Structural analysis revealed that four E104 Fabs can simultaneously bind to one tetrameric β-tryptase via an exosite epitope. Thus, E104 can be classified as an indirect exosite binder similar to other Fabs that bind proteases such as HGFA, hepsin, BACE1, and MMP9[29–31]. This allosteric site is very similar to that recognized by the 31A.v11 antibody[32], except that E104.v1 engages the epitope at a slightly different angle and fails to induce significant conformational changes in both tetramer interfaces simultaneously (Fig. 2e and Supplementary Fig. 8). This shift for E104 moves CDRL1 away from the 60's loop and prevents conformational changes induced by CDRL1 of 31A.v11, which leads to disruption of the large interface (blue circle in Fig. 2e). This is consistent with the finding that neither E104.v1 Fab nor E104.v2 Fab can destabilize the large interface as shown by the SEC analysis (Fig. 3c). However, the different angle of binding allows four E104 Fabs to simultaneously engage with a single tryptase tetramer, whereas steric clashes between 31A.v11 Fabs when bound to the tetramer prevent this possibility, which is the basis for its MOA[32].

We used HDX-MS analysis to compare the binding of v1 and v2 to tryptase in solution and found that a subset of residues (Q81-L83) were only protected from H-D exchange upon binding to v2, but not to v1. These residues are within 4 Å of CDRH2 in the E104.v1 Fab (violet spheres in Fig. 2f) and comprise a portion of a loop that contains Y74 and Y75, which are key contact residues in the small interface. E104.v1 and E104.v2 differ at key Vernier residues (V71R and F78V) and R71 strongly impacts the conformation of CDRH2[41,42]. Existing literature indicates that that the R71 in E104.v2 forms a hydrogen bond with S53 in CDRH2, which might push the tip of CDRH2 (S54/A55) into a steric clash with Q81-L83 in β-tryptase. The conformational changes we observed in this region and the neighboring 70's loop that includes Y74 and Y75 thus likely explains why E104.v2 can destabilize the small interface while E104.v1 cannot, despite possessing identical CDR sequences. This effect was confirmed with SEC analysis using WT and disulfide-locked β-tryptase variants (Fig. 3c).

Our most striking observation is that E104 in the presence of heparin is a potent tryptase inhibitor as a bivalent IgG, but loses nearly all inhibitory activity when reformatted to a monovalent Fab. Additionally, a bivalent IgG alone is not sufficient for inhibition since the IgG2 isotype is also inactive. To probe the molecular requirements, we constructed a diverse panel of IgG hinge variants with different length, rigidity, and constraint (Table 1). We found that full inhibitory activity required a sweet spot in upper hinge lengths of 8-12 residues. The epitope spacing and hinge properties combine to generate a scenario in which simultaneous binding of two Fabs from one IgG can occur and the steric strain induced by this binding is sufficient to pry apart the protomers in the tetramer into inactive monomers. Indeed, tethering the two Fab arms together via an engineered cysteine or as an IgG2 resulted in a loss of inhibitory activity. Steric strain between the Fab arms has recently been observed to enable IgGs to walk along surfaces with regularly patterned epitopes[43]. In this case, bivalent binding to antigens coated at a regular spacing of 60 Å induced a steric strain between the two Fab arms, which led to movement or walking of anti-Strep-Tag II, anti-Sendai virus and anti-rhinovirus IgGs across the surface. In our case, the similar spacing of the epitopes on tetrameric tryptase results in the steric strain upon bivalent binding, which leads to dissociation of the tetramer. Future studies aimed at measuring the energetics of this physical strain would be both informative and interesting.

Heparin is known to stabilize the tryptase tetramer via the small interface, as well as allosterically condition the active site[10,17,18]. We found this stabilizing effect is also sufficient to prevent the E104.v2 Fab from promoting dissociation of the tryptase tetramer, thus preventing the Fab from having any significant inhibitory activity. In contrast, the IgG retains potent activity in the presence of heparin (Fig. 3i). As a whole, these data indicate that the predominant MOA is due to bivalent binding to neighboring exosites and subsequent steric strain induced tetramer dissociation. Furthermore, the inhibitory activity of E104.v1 IgG (Fig. 1b), which lacks any allosteric effect on the small interface, provides additional support that the molecular pliers MOA dominates over most IgG:β-tryptase ratios. This dissociation may occur via the large interface or via some other allosteric mechanism. Existing data indicates that heparin serves as a stabilizing factor for two dimeric tryptases that are bound via the large interface into the final tetramer and heparin by itself cannot sufficiently stabilize the small interface to permit two monomers assemble into an active stable dimer[10,11]. Therefore, disrupting the large interface would result in a pair of small interface bound dimers that likely rapidly fall apart into inactive monomers.

Previous efforts to study the role of the hinge in antibody activity have demonstrated that the hinge can regulate effector function. For example, altering the hinge length or rigidity changes the interactions with C1q or FcγRs and impact the complement-dependent cytotoxicity (CDC) or antibody-dependent cellular cytotoxicity (ADCC) activity of the antibody, respectively[44]. Another general antibody feature controlled by the hinge is IgG wingspan. While previous efforts[35,36] with immune complexes and SAXS analysis have demonstrated that different IgG isotypes can exhibit different degrees of flexibility between the two Fabs arms, few examples exist demonstrating this flexibility directly contributes to antibody function[45,46]. In one case, engineering the longer and more flexible IgG3 upper hinge in place of the IgG4 upper hinge improved the agonist activity of two anti-thrombopoietin receptor mAbs by ~10-fold[45]. The concept of hinge-dependent function can also be applied to bispecific antibodies. In this case, one can design function at two epitopes on the same protein or at two distinct proteins. For example, the activity of a bispecific antibody designed to mimic factor VIII by binding to Factor IX and Factor X could be modulated by choice of the IgG isotype and rigidity of the upper hinge[46]. In summary, our findings suggest that engineering the length and rigidity of the upper hinge may provide a route to engineering agonist and antagonist antibodies.

## Methods

**Recombinant expression of WT and mutant β-tryptase variants.** The sequence encoding mature wild-type (WT) human βI-tryptase (Uniprot Q15661) from Ile31-Pro275 (Ile16-Pro246 chymotrypsinogen numbering) was cloned into a modified pAcGP67A vector behind the polyhedron promoter and the gp67 secretion signal sequence. Expression constructs contain an N-terminal His$_6$-tag directly followed by an enterokinase (EK) cleavage site directly fused to the mature N-terminal Ile16 of β-tryptase. Site-directed mutagenesis was performed using standard Quik-Change protocols (Agilent, Santa Clara CA) to generate β-tryptase mutants. All constructs were confirmed by DNA sequencing. Recombinant baculovirus were generated using the Baculogold system (BD Biosciences, San Jose CA) in Sf9 cells following standard protocols. Trichoplusia ni cells were infected for large-scale protein production and harvested 48 h post-infection. The harvested media was supplemented with 1 mM NiCl$_2$, 5 mM CaCl$_2$ and 20 mM Tris pH 8, shaken for 30 min and then centrifuged for 20 min at 8500 × g to remove the cells and precipitate from media. The supernatant media was filtered through a 0.22 μm PES filter prior to loading onto a Ni-NTA affinity column.

Insect cell media containing secreted His6-tagged zymogen β-tryptase (WT or mutant) was loaded onto a 10 mL Ni-NTA Superflow column (Qiagen, Germantown, MD) at a volumetric flow rate of 170 cm/h. The column was washed with 10 column volumes (CV) of wash buffer (20 mM Tris pH 8, 10 mM imidazole, 300 mM NaCl) and eluted with 8 CV elution buffer (20 mM Tris pH 8, 300 mM imidazole, 300 mM NaCl). Fractions assayed by SDS-PAGE containing β-tryptase were pooled, concentrated and loaded onto an S200 column (GE Healthcare, Piscataway, NJ) for further purification by size-exclusion chromatography (SEC) using SEC buffer (10 mM MOPS pH 6.8, 2 M NaCl) at flow rates recommended by manufacturer. Fractions containing zymogen β-tryptase (monomeric) were pooled and concentrated. Zymogen β-tryptase was then cleaved overnight at room temperature at a concentration of 2 mg/mL in 10 mM MOPS pH 6.8, 0.2 M NaCl containing 0.5 mg/mL heparin (H3393; average MW ~ 18 kDa; Sigma Aldrich, St. Louis, MO) and 0.1 mg/mL EK (NEB, Ipswich, MA). This step removes the N-terminal His6-tag and results in tetramerization and proteolytically active β-tryptase, which has IVGG as the newly formed N-terminal sequence starting at residue 16. Tetrameric β-tryptase was then subjected to SEC using an S200 column (GE Healthcare, Piscataway, NJ) in SEC buffer to purify tetrameric β-tryptase by removing EK and any uncleaved zymogen β-tryptase. β-tryptase mutants Y75C and I99C were purified by Ni-affinity chromatography as described above. Disulfide-linked β-tryptase dimer mutants were then separated from non-disulfide-linked β-tryptase monomer mutants by SEC on an S200 column as above. Disulfide-linked dimer mutants were further processed by EK cleavage as described above for WT β-tryptase to form active tetramers (mutants Y75C and I99C).

**Humanization of E104 antibody.** The VL and VH domains from the rabbit E104 were aligned with the human VL kappa I (VL$_{KI}$) and human VH subgroup IV (VH$_{IV}$) consensus sequences. The hypervariable regions (HVR) were engineered into the consensus human VL$_{KI}$ and VH$_{IV}$ acceptor frameworks to generate CDR-graft variants. To evaluate framework Vernier positions that might be important, selected Vernier positions were mutated back to the rabbit sequences. The Vernier positions include 2, 4, 43, 68, and 87 in VL and 37, 67, 71, 78, and 91 in VH. In total, two different versions of humanized VL sequences and six different versions of humanized VH sequences were synthesized and subsequently subcloned into mammalian expression vectors (Genewiz, South San Francisco, CA). By combining the different versions of LC with HC, a total of twelve different humanized E104 variants (v1 to v12) were generated.

**Generation and purification of Fab fragments.** Fabs were cloned and expressed in E. coli[47,48]. Briefly, E.coli cell paste containing the expressed Fab was harvested from fermentations and dissolved in phosphate-buffered saline (PBS) buffer containing 25 mM EDTA and 1 mM phenylmethylsulfonyl fluoride. The mixture was homogenized and then passed twice through a microfluidizer. The suspension was then centrifuged at 21,500 × g for 60 min. The supernatant was then loaded onto a Protein G column (GE Healthcare, Piscataway, NJ) equilibrated with PBS at 5 mL/min. The column was washed with PBS buffer and proteins were then eluted with 0.6% acetic acid. Fractions containing Fabs were pooled and then loaded onto a 50-mL SP Sepharose column (GE Healthcare, Piscataway, NJ) equilibrated in 20 mM MES pH 5.5. The column was washed with 20 mM MES buffer pH 5.5 for 2 column volumes and then eluted with a linear gradient to 0.5 M NaCl in 20 mM MES buffer pH 5.5. For final purification, Fab-containing fractions from the ion exchange chromatography were concentrated and run on a S75 size-exclusion column (GE Healthcare, Piscataway, NJ) in PBS buffer.

**Generation and purification of anti-tryptase antibodies**. All the humanized and engineered anti-tryptase variants were expressed by transient transfection of 293 cells. Antibodies were purified by affinity chromatography and SEC using standard methods (MabSelect SuRe; GE Healthcare, Piscataway, NJ).

**Surface plasmon resonance experiments**. The affinity of each E104 variant for human $\beta$I-tryptase was determined by surface plasmon resonance using a Biacore T200. Biacore Series S CM5 sensor chips were immobilized with 150–400 response units (RUs) of monoclonal mouse anti-human IgG (Fc) antibody (Human antibody capture kit; GE Healthcare, Piscataway, NJ) and anti-tryptase variants were subsequently captured on each flow cell. Serial threefold dilutions of the human $\beta$I-tryptase monomer were injected at a flow rate of 30 $\mu$L/min at 25 °C. Each sample was analyzed with 3-min association and 10-min dissociation. After each injection the chip was regenerated using 3 M MgCl$_2$. Binding response was corrected by subtracting the RU from a flow cell capturing a control IgG at a similar density. A 1:1 Langmuir model of simultaneous fitting of association ($k_{on}$) and dissociation ($k_{off}$) rate constants was used for kinetic analyses.

**$\beta$-tryptase enzymatic activity assay**. The enzymatic activity of 1 nM tetrameric $\beta$I-tryptase was measured in 200 $\mu$L volume using 2 mM S-2288 chromogenic substrate (DiaPharma Group, Inc., West Chester, OH) was performed as previously described[11]. The reaction velocity was determined by the rate at which p-nitroaniline (pNA) is released, which was measured spectrophotometrically at 405 nm using a SpectraMax M5e plate reader using Softmax Pro (v6.2.2) (Molecular Devices, Sunnyvale, CA). Inhibition plots were generated using 4-parameter fits from data collected from three independent experiments and statistical analysis was performed using KaleidaGraph (v4.1.3) (Synergy Software, Reading, PA).

**Bronchial smooth muscle cell assay culture conditions**. Human bronchial smooth muscle cells (BSMC) (Lonza, Walkersville, MD) were cultured in a humidified incubator at 37 °C with 5% CO$_2$ in complete culture media SmGM-2 (Lonza, Walkersville, MD). Assay media is SmGm-2 culture media without human serum or supplements added (Lonza, Walkersville, MD).

**Bronchial smooth muscle cell proliferation assay**. One day prior to performing a proliferation assay, BSMC were plated at $2 \times 10^5$ cells/mL in a 96-well tissue culture plate (Falcon BD, Franklin Lakes, NJ) in complete culture media. After 24 h, culture media was replaced with assay media and cells were incubated for an additional 24 h. Then, E104.v2 IgG4 was serially diluted 3.3-fold in assay media in a 96-well tissue culture plate (Falcon BD, Franklin Lakes, NJ). One-hundred microliters of E104.v2 IgG4 was transferred to a 96-well plate containing 100 $\mu$L of 200 nM human $\beta$I-tryptase. Tryptase and E104.v2 IgG4 were incubated for 30 min at 25 °C. The assay media was removed from the plated cells and replaced with 100 $\mu$L of the diluted antibodies plus $\beta$I-tryptase. The final concentration of antibodies ranged from 2.0 $\mu$M to 4 pM. The final concentration of $\beta$I-tryptase was 100 nM. Wells with $\beta$I-tryptase alone were included as stimulation controls. Wells with assay media alone were included as unstimulated controls. Plates were incubated for 24 h in a 37 °C before the addition of 1 $\mu$Ci of $^3$H-thymidine (PerkinElmer, Waltham, MA) per well. After an additional 6 h of incubation, proliferation was measured by $^3$H-thymidine incorporation. Cell-associated radioactivity was quantified by scintillation counting. Results are expressed as the mean of triplicate samples. Graphs were generated and statistical analysis was performed using KaleidaGraph (v4.1.3) (Synergy Software, Reading, PA).

**Bronchial smooth muscle cell contraction assay**. One day prior to performing the cell contraction assay, BSMC were plated in collagen at $9 \times 10^6$ cells/mL in a 24-well plate (Falcon BD, Franklin Lakes, NJ) following the manufacturer's guidelines (Cell BioLabs Inc., San Diego, CA). After a 1-h incubation at 37 °C, cells were overlaid with 1 mL of assay media. After 24 h, media at 37 °C, media was replaced with 250 $\mu$L of fresh assay media. $\beta$-tryptase was diluted in 250 $\mu$L assay media to 660 nM in the presence or absence of 4 $\mu$M of E104.v2 IgG4 and incubated for 30 min at 37 °C prior to addition to the specified wells containing the cell: collagen matrix. The final concentrations were 330 nM $\beta$-tryptase and 2 $\mu$M antibody. Cell contraction was initiated by the release of the cell:collagen matrix from the plate wall using a sterile pipette tip. Assay media alone was used as an unstimulated control. At the start of cell contraction ($t = 0$), cell:collagen matrices were visualized, imaged and recorded using a Protein Simple AlphaI Imager. Cells were incubated at 37 °C for an additional 3 h and cells:collagen matrices were reimaged and recorded ($t = 3$). Data was analyzed using NIH Image J (v1.48). Data is represented as the percent change in cell:collagen matrix diameter from the start of contraction ($t = 0$) until the 3 h timepoint ($t = 3$). Results are expressed as the mean of triplicate samples.

**Electron microscopy analysis of E104.v1 Fab and $\beta$I-tryptase**. Four microliters of purified $\beta$I-tryptase bound to E104.v1 Fab fragment were applied to a freshly glow discharged 400-mesh copper EM grid covered with a thin layer of continuous carbon (Ted Pella, Redding, CA). After 30 s of incubation, the grids were stained with 5 drops of a freshly prepared 0.75% (w/v) uranyl formate solution. Negatively

stained EM grids were imaged on a Tecnai T12 microscope (Thermofisher, Waltham, MA) operated at 120 kV with a magnification of x67,000 (2.02 Å/pixel at the detector level) using a defocus range of −0.8 to −1.3 μm. Images were recorded using a 4k x 4k CCD camera (UltraScan 4000, Gatan Inc., Pleasanton, CA).

Particle picking for all datasets was executed using Eman2 e2boxer v2.1[49] and a 200 x 200-pixel particle box size window. Reference Free 2D class averaging of individual complexes was generated using iterative Multivariate Statistical Analysis (MSA) and Multi-Reference Alignment (MRA) in IMAGIC v10[50]. 3D structure determination for both complexes was obtained using Relion v3[51]. For tetrameric $\beta$I-tryptase bound to E104.v1 Fab we used, as initial model, a 3D map generated from the $\beta$-tryptase tetramer crystal structure (PDB: 4A6L), using pdb2mrc in Eman v2.1 and filtered to a final resolution of 60 Å. 3D reconstruction of $\beta$I-tryptase:E104.v1 Fab complexes were visualized using Chimera v1.12[52]. Crystal structures of tetrameric $\beta$I-tryptase and Fab models, generated using with Modeller v9.22 and Herceptin Fab (PDB:1N8Z) as a starting model, were fitted into the EM 3D maps using the fit in map algorithm in Chimera v1.12[52].

**Complex formation of $\beta$I-tryptase tetramers with Fabs**. Tetrameric $\beta$I-tryptase, wild-type or mutants Y75C or I99C, were mixed with a twofold molar excess of Fab and incubated for 15 min at 25 °C. The mixture was then subjected to S200GL size-exclusion column (25 mL bed volume, GE Healthcare, Piscataway, NJ) chromatography in either tryptase SEC buffer (10 mM MOPS pH 6.8, 2 M NaCl) or TNH (20 mM Tris pH 8.0, 150 mM NaCl, 0.1 mg/mL Heparin) buffer. All fractions containing protein as indicated by absorbance at A$_{280\ nm}$ were analyzed by SDS-PAGE under reducing conditions with $\beta$-mercaptoethanol to identify the protein components in these fractions. SeeBlue$^{TM}$ Plus2 pre-stained protein standard (ThermoFisher, Waltham, MA) was used for molecular weight markers for SDS-PAGE analysis.

**X-ray crystallography of E104 Fab and tetrameric $\beta$I-tryptase**. Crystals of E104.v1 Fab with $\beta$I-tryptase were grown at 19 °C using vapor diffusion by mixing protein in 1:1 (v/v) with a reservoir solution containing 0.1 M Tris pH 8.5, 0.2 M CaCl$_2$, 20% PEG 4000 and 4% pentaerythritol ethoxylate. The crystals were cryo-protected in artificial mother liquor containing 0.1 M Tris pH 8.5, 0.2 M CaCl$_2$, 35% PEG3350 and preserved for data collection by sudden immersion in liquid nitrogen. Diffraction data were collected at SSRL beamline 12-2 in a monoclinic lattice extending to 3 Å resolution using a Pilatus 6 M pixel array detector and 0.9795 Å wavelength X-rays. Data were reduced using XDS v1.1.5[53] and scaled with CCP4 v6.1[54], and the structure solved by molecular replacement using Phaser v2.5.5[55] in space group P2$_1$ revealing a $\beta$I-tryptase tetramer bound by four Fabs. The molecular replacement search probes were a $\beta$-tryptase protomer from PDB accession 4A6L and an antibody Fab fragment derived from PDB accession 1FVD by scanning modified versions with a range of elbow angles using the rotation function only. After limited refinement, one Fab constant region was replaced in a molecular replacement search using just the constant region from 1FVD. Chymotrypsinogen numbering was used for $\beta$-tryptase and the Kabat numbering was used for the E104.v1 Fab. Model and electron density map inspection and adjustments were performed using Coot v0.8.6[56] and the structure refined using Buster v2.11.6[57]. Carbohydrate was not built into suggestive but poor electron density for $\beta$I-tryptase residue Asn222. Data collection and refinement metrics appear in Table 2. The Ramachandran plot showed that 94% of residues are in the most favored regions with no residues in disallowed regions. Figures with structures generated by X-ray crystallography were made using PyMOL v2.2[58].

**Hydrogen deuterium exchange mass spectrometry analysis**. Deuterium uptake rates of monomeric $\beta$I-tryptase in the presence and absence of antibody E104.v1 and E104.v2 were measured to determine structural regions that are modified upon antibody binding. Bound samples contained 1:1 mixture of $\beta$I-tryptase and the respective antibody (98% bound using measured dissociation constants) prepared and incubated at 25 °C for 1 h. $\beta$I-tryptase concentration prior to deuterium labeling was 30 $\mu$M in Ab bound and unbound samples. HDX-MS experiments consisted of diluting samples 15-fold into deuterium labeling buffer containing 20 mM histidine acetate at pD 7.0 (Serva Electrophoresis GmbH, Heidelberg, Germany) at 20 °C. Six labeling times (0.5, 2.3, 10, 48, 218, 1000 min) were sampled in duplicate, quenched by lowering the pH to pH 2.5 and the addition of 2 M guanidinium chloride (Sigma Aldrich, St. Louis, MO) and 0.25 M TCEP (Sigma Aldrich, St. Louis, MO) and injected into a cold online system using the LEAP XT-Pal robot (Trajan Scientific, Austin TX). Samples were first passed through an immobilized pepsin column (2.1 × 30 mm, Applied Biosystems, Foster City, CA) and loaded onto an online trap column (Acquity Vanguard C$_8$, Waters) for desalting at 150 $\mu$L/min for 2 min at 0 °C. Peptide fragments were then separated by reversed-phased chromatography (50 $\mu$L/min) using an Acquity UPLC BEH C$_{18}$ (1.7 $\mu$m particle size, 1.0 × 50 mm) (Waters, Milford, MA) and introduced into the mass spectrometer (Thermo Orbitrap Elite, 120 kHz resolution at $m/z$ 400) for mass analysis. To further minimize back-exchange, the ion transfer tube temperature was set to 175 °C, and 0.04% v/v TFA was added to mobile phases containing 0.1% formic acid to adjust their pH approximately to 2.25. The levels of recovery on average are thus expected to be roughly 85%[59].

Mascot v2.1 was used to search MS/MS runs, MS1 and 2 search tolerances were 15 ppm and 0.8 Da error, respectively, the minimum peptide size was four residues (no max), and the search database contained only sequences for E104 and tryptase as dedicated consumable equipment was used for these measurements. Results with Mascot scores <35 are removed as these represent less confident identifications. To expand the size of the peptide pool and produce many overlapping peptides, all peptides identified by MS/MS were added to an exclusion list in the mass spectrometer for the entire duration of a second MS/MS experiment[60]. Peptides identified were pooled together and used as a reference to find deuterated the labeling experiments. Default Thermo settings are used in MSconvertGUI v3.0.5741 to convert raw files into the open source mzXml format before analysis. The ExMS program v1[61,62] was used to identify deuterated peptides and prepare extracted ion chromatograms, which were then analyzed for deuterium content using in-house implementations of a n-population binomial analysis using a statistical F-test exactly described in chapter 4.4 and appendix A[63]. Prior to fitting isotopic distributions, degenerate charge states were combined by direct summation of equivalent peak intensities. Statistical parameters relevant to peptides the experiments are given in Supplementary Data 1; note that degenerate peptides (those with multiple charge states) were only counted once in determination of experimental redundancy.

Regions of notably different deuterium uptake upon binding to each Fab were those defined as having a mean difference in deuterium uptake that exceeded the sum of the range in uptake for both apo and holo randomized duplicate measurements (for any timepoint). Overlapping peptides were manually compared and used to define residues shown in Supplementary Table 2 and colored onto Fig. 2f. All peptides used in this work along with error bars are shown in Supplementary Fig. 9 (E104.v1) and Supplementary Fig. 10 (E104.v2). A more quantitatively rigorous analysis, using empirical protection factors (PFs)[59], defines how to compute an uncertainty estimate in empirical PFs and shows that quantitatively significant differences between Fabs are in general agreement with the manual analysis, once accounting for the propagation of uncertainty quantitatively (Supplementary Note 1). Therefore, we defaulted to manually annotated data. A summary of the HDX data is provided as Supplementary Data 1 and Supplementary Data 2 is now included containing every relevant calculation and may be of particular interest to practitioners of HX MS experiments.

**Reporting Summary**. Further information on research design is available in the Nature Research Reporting Summary linked to this article.

## Data availability
An electron microscopy density map for the negative staining reconstruction of βI-tryptase tetramer bound to E104.v1 Fab has been deposited in the Electron Microscopy Data Bank under accession code EMD-21389. The crystal structure of the βI-tryptase tetramer bound to E104.v1 Fab has been deposited in the protein databank (PDB) with PDB accession code 6VVU. HDX-MS data for E104.v1 and E104.v2 epitope mapping has been deposited in the MassIVE Repository with accession codes MSV000086368 and MSV000086370. Source data are provided with this paper.

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

## Acknowledgements

We thank Menno van Lookeren Campagne, Rick Brown, Yan Wu, and Elaine Chang for their helpful input on the project. We thank the antibody production group and bio-molecular resource group for respective antibody and tryptase expression and purification. We thank Erin Christensen and Julie Hang for purification of tryptase and preliminary SEC experiments. We thank Mark Ultsch for submitting X-ray coordinates to the PDB and preparation of Supplementary Fig. 6. Use of the Stanford Synchrotron Radiation Lightsource, SLAC National Accelerator Laboratory, is supported by the U.S. Department of Energy, Office of Science, Office of Basic Energy Sciences under Contract No. DE-AC02-76SF00515. The SSRL Structural Molecular Biology Program is supported by the DOE Office of Biological and Environmental Research, and by the National Institutes of Health, National Institute of General Medical Sciences (P41GM103393). The contents of this publication are solely the responsibility of the authors and do not necessarily represent the official views of NIGMS or NIH.

## Author contributions

H.R.M., T.Y., R.A.L., and J.T.K. conceived the project and designed experimental approaches. M.T.L. and D.K. generated and characterized rabbit antibodies. H.R.M. performed SEC experiments and activity assays and purified and characterized the complexes for crystallography. J.K.J., A.M., and K.M.L. performed the in vitro characterization and cell-based assays. P.W. also generated crystals and C.E. solved the crystal structure. A.E. generated grids and C.C. analyzed the EM data. B.T.W. performed and analyzed the HDX-MS data. X.C., M.S.D., R.V., and J.T.K. carried out the antibody engineering and binding kinetics. Y.F. generated tryptase expression constructs. H.R.M., R.A.L., and J.T.K. interpreted the data and wrote the manuscript with input from all authors. R.A.L. and J.T.K. are co-senior authors and supervised the project.

## Competing interests

All authors were employees and/or shareholders of Genentech Inc. at the time the work was done.
