## [Peer Review File · Nature Communications]

Reviewers' comments:

Reviewer #1 (Remarks to the Author):

Maun et al. describe work developing a novel antibody-based inhibitor to human β -tryptase, a serine protease enzyme that is implicated in allergic inflammatory responses. β -tryptase has long been the target of small molecule drug development, but progress has been frustrated by numerous technical hurdles. Further, the ternary structure of the homo-tetrameric molecule has limited approaches using endogenous protease inhibitors. With these issues in mind, the authors sought to generate a potent β -tryptase inhibitor using an antibody approach. They were successful in this endeavor resulting in a potent antibody (E104), which was humanized by varying key scaffold residue sites. From the cohort of 12 humanized variants, the two best binding antibodies were selected for further characterization. Interestingly, these two antibodies: E104.v1 and E104.v2 differed only in two Vernier framework residues, V71R and F78V. Curiously, these framework residues led to quite different inhibition properties comparing v1 to v2. Comprehensive structure and functional data were collected and analyzed with the conclusion that apparent subtle changes in the framework of an antibody's antigen binding site can produce rather profound differences in their apparent mechanisms of action (MOA). Interestingly, even though v1 and v2 share virtually the same epitope, the changes in the framework of v2 appear to allosterically induce conformational changes in β -tryptase that significantly affect the tetrameric assembly of the enzyme, which is demonstrated by H/D-mass spec exchange data. Notably, the dissociation of the tetramer by v2 can be partially ameliorated by the addition of heparin.

To establish whether the heparin had similar effects in the context of other IgG formats, the Fab v2 was reformatted into IgG1, IgG2 and IgG4 isotypes. The authors determined that IgG1 and IgG4 were potent inhibitors, but IgG2 was not. To sort out what was happening, a panel of linker arm hinge mutants was constructed that were designed to alter hinge length and flexibility. This systematic search allowed the authors to hone-in on a sweet spot of hinge length and composition that maximized the effective inhibition of the antibody. From these data, the authors hypothesize that binding of the two Fab arms of v2 to the two diagonally dispose subunits of tryptase, when presented in an optimized IgG scaffold construction, effectively induces conformational strain that results in disassembly of the enzyme's tetrameric ternary structure.

Overall, this manuscript describes a high quality piece of research that should elicit strong interest from the antibody engineering community, as well as the general reader. Below are comments that the authors might consider.

- 1). The differences in the antibody binding effects to β -tryptase with and without the stabilization by heparin are a big piece of the story. How much does heparin actually stabilize the tetramer? A simple DSF experiment could clarify this. Once dissociated, can the monomers reassociate into a functional tetramer?

2). A significant amount of work was done systematically modifying the linker regions to produce a series of IgG variants. It was surprising to me that the structural information that was available from the Fab v1- β -tryptase complex was not better utilized to guide and explain the linker engineering.

At line 454 begins a discussion about the length of linker arm (defined by the P217-C226 distance) of an IgG. Maybe I'm missing something, but the Fab v1 complex structure should provide much more insight into what is taking place in the case of IgG v2 than relying on IgG1 (PDB 1HZH). The Fab v1 has to have virtually the same epitope as v2 and thus, even taking into consideration the authors' suggestion that engagement angles might differ, the v1 structure should provide a good model for the orientation of the Fab. This, in turn, should allow a determination of the directionality and pretty much the point where the beginning part of the linker (P217) exits the heavy chain constant domain. The coordinates are not released yet and therefore it is not possible to determine how far the chain can be traced to P217; nevertheless, any approximations based on a relevant structure should provide more reliable information to build from than a generic IgG structure.

4). Molecular pliers- This conceptual model of the MOA of the inhibitory effect of E104.v2 is built on the premise that coupled allosteric effects together with steric stress generated by this antibody "pry" the tetramer apart. There seems to be two parts here. Fab v1 dissociates the tetramer, heparin rescues it. It is probably possible to back out the energies associated with the Fab induce dissociation, but assigning the energy contributed by the "stress" induced by the pliers part is harder to get at experimentally. This is more a comment than criticism, but it would be interesting to work out the contribution of the Fab v1 alone to put the pliers' contribution in better context.

3). Table 1 reports the KDs of the hinge sequence variants. The caption suggests that the affinities are to β -tryptase, which one might assume relates to the tetramer. However, the SPR data were collected on the monomer, so this should be clarified. The accuracy/precision of the SPR data also seem to be overinterpreted. There is common agreement among biophysicists that the accuracy of the numbers spit out from SPR sensograms fitting programs have to be taken with a grain of salt for ultra-high affinity interactions. To suggest that KD differences in the 0.5 nM range can be measured to an accuracy to the 2nd decimal place is pure fantasy. The supplementary material should include the kinetics and the sensograms for at least 5 of the best variants. The dissociation data were collected on 3 serial dilutions over 600 secs. It would take additional serial dilutions and require considerably longer collection times to generate KD accuracies that would be statistically relevant to the ones reported here. All that said, determining and reporting binding constants to that level of accuracy does not add much to the conclusions. The authors should use their judgement about how to report these data to ensure that the small differences in affinity between the variants are not over interpreted.

Reviewer #2 (Remarks to the Author):

The manuscript entitled " Bivalent Antibody Pliers: Inhibition of β -tryptase by an allosteric mechanism 1 dependent on the IgG hinge" represent a comprehensive functional and structural study that aims at deciphering the mechanism of action of antibodies targeting β -tryptase. It is a nice example of combining high resolution techniques such X-ray and EM giving rather static picture of each individuals and low resolution approaches like HDX-MS answering dynamic properties of the complex.

There are major issues that must be addressed regarding HDX-MS data reporting and major revision is required.

1) HDX-MS experiments were performed using a 1:1 mixture of β I-tryptase:Fab (incubated at 25°C for 1 h) while SEC analyses were performed at 1:2 β I-tryptase:Fab (incubated for 15 min at 25°C). Why not using the same ratio for these two experiments? What is the quality control of the HDX-MS sample? Any SEC or native MS to probe that most of the sample correspond to β I-tryptase:Fab complex, without extensive free β I-tryptase or Fab that would avoid proper HDX data interpretation.

2) It would have been interesting to have online SEC-native MS measurements for direct identification of SEC species, more informative than 1D SDS-PAGE gel.

3) References related to HDX-MS experiments are rather old and more generally HDX-MS data do not fulfil current recommendation of the HDX-MS community published in Nat Methods. 2019; 16(7): 595–602. I would suggest authors to update their material and method section and supplementary data, among which

- temperature at which the H/D reaction was performed
- pertinent LC-MS settings (for example, the LC gradient and flow rates, reversed-phase columns used, MS ion source parameters etc.) should be reported
- duration of the digestion, flow rate and the temperature that the digestion was conducted under.
- detail of the time points
- Please include the search parameters used for peptide identification (sequence database used for peptide search, MS1 and MS2 tolerance, min and max peptide length, fixed and variable modifications, cleavage sites and number of missed cleavages if applicable, score and FDR thresholds)
- threshold for significant differences in HDX (a threshold value interpreted as representing a significant difference in HDX between examined protein states based on the quantitative measure of repeatability. It is also beneficial to consider applying objective statistical analysis to bolster authors conclusions.

- When reporting explicitly on the change of HDX in a peptide, it is usual to present peptide uptake plots, plotting each labeling time with the per-peptide standard deviation. It might also give better overview for the reader to show H/D plots for all peptides. Authors suggest some rearrangement of the structure during the complex formation and showing the entire H/D data will shed a light on the protein dynamic not only selected part.
- Please explicitly state the mapping methodology on the X-ray 3D structure and at which time points data are depicted, based on a quantitative and statistical argument applied to the entire dataset.
- Precise if HDX data were corrected for back exchange or not.

Similarly to X-Ray data, such HDX-MS data could be provided as supplementary materials. Examples of templates could be uploaded as supplementary files from Nat Methods. 2019; 16(7): 595–602

Reviewer #3 (Remarks to the Author):

Review report Maun et al 2020

The manuscript by Maun et al. is a beautiful piece of molecular analysis of antibodies directed against the human mast cell tryptase. The manuscript is close follow up of their recently published also equally brilliant article in Cell. The focus of the article is primarily the use of a large set of hinge variants of a humanized monoclonal that targets a region in the interphase between the four subunits of the human mast cell tryptase where they show that divalent binding in combination with distance constrains are of major importance for the inactivation of the tryptase by pushing the subunits apart.

I have few if any comments on the actual work which is a beautiful piece of molecular design showing the importance of the hinge for the function of various human IgG isotypes.

My major concern is at another level and that the background information. The authors almost entirely neglect the presence of the other mast cell granule proteases that are stored together with the tryptase and their effects. I think they need to mention both the chymase, the mast cell specific carboxypeptidase CPA3 and cathepsin G and their role in mast cell function as well as potential side effects of an anti-tryptase antibody.

It has recently been shown that the human mast cell chymase and the tryptase may act together to control excessive TH2 mediated immunity by cleaving a highly selected set of cytokines and chemokines. They almost exclusively cleave five potent TH2 cytokines out of a very large panel of cytokines and chemokines, including three of the most well known TH2 cytokines, IL-33, IL-18 and TSLP and also two additional that recently have been shown to act as potent TH2 inducing cytokines namely IL-15 and IL-21.

Knock out of the mouse counterpart of the human mast cell chymase mMCP-4 has also been shown to enhance sensitization giving further support of this potential effect of these proteases.

I understand that Genentech is a commercial company pushed by investors to come up with new drugs but when publishing in respectable international journals it is important also to include potential side effects of their efforts. I therefore suggest the authors also to include information stating the presence of tryptase as one out several mast cell specific or related proteases and that these act together upon mast cell activation and that there are potential side effects with long term treatment with anti-tryptase and/or anti chymase antibodies potentially could affect TH1/TH2 balance and actually increase sensitization to earlier allergen and potentially also initiate novel sensitization. Their cleavage of a number of neuropeptides and other peptide hormones and various toxins (Chymase and CPA3) may also be factors that are of importance when looking at ways to inhibit their activity.

It is a beautiful piece of work that however needs to be put into a broader context. There is most likely a reason why both the chymase and the tryptase have been maintained in all mammalian lineages for what it appears more than 200-250 million years of mammalian evolution. Chymases are present in all three extant mammalian lineages with very similar cleavage specificities. The so far only exceptions are rabbit and guinea pig where their chymases have become strict Leu-ases and not classical chymases. The picture of the tryptases is less complete but the genes are present in all mammalian lineages although any detailed analysis have been performed on marsupial and monotreme tryptases. The conservation of both enzymes gives strong indications for evolutionary important and therefore conserved functions. Long term inhibition of these components of our immune system may thereby involve certain amount of risk, which I think needs to be addressed to give a more correct view of potential side effects of anti tryptase antibody treatment.

The low level or absence of immediate side effects after injection of the anti tryptase antibodies make the analysis of their activity highly interesting. However, during such studies of their potential beneficial clinical effects, potential long term negative effects should also be kept in mind not to put patients involved in such clinical tests at risk. Questions that I think needs to be addressed in the article.

Reviewer #1:

Maun et al. describe work developing a novel antibody-based inhibitor to human β -tryptase, a serine protease enzyme that is implicated in allergic inflammatory responses. β -tryptase has long been the target of small molecule drug development, but progress has been frustrated by numerous technical hurdles. Further, the ternary structure of the homo-tetrameric molecule has limited approaches using endogenous protease inhibitors. With these issues in mind, the authors sought to generate a potent β -tryptase inhibitor using an antibody approach. They were successful in this endeavor resulting in a potent antibody (E104), which was humanized by varying key scaffold residue sites. From the cohort of 12 humanized variants, the two best binding antibodies were selected for further characterization. Interestingly, these two antibodies: E104.v1 and E104.v2 differed only in two Vernier framework residues, V71R and F78V. Curiously, these framework residues led to quite different inhibition properties comparing v1 to v2. Comprehensive structure and functional data were collected and analyzed with the conclusion that apparent subtle changes in the framework of an antibody's antigen binding site can produce rather profound differences in their apparent mechanisms of action (MOA). Interestingly, even though v1 and v2 share virtually the same epitope, the changes in the framework of v2 appear to allosterically induce conformational changes in β -tryptase that significantly affect the tetrameric assembly of the enzyme, which is demonstrated by H/D-mass spec exchange data. Notably, the dissociation of the tetramer by v2 can be partially ameliorated by the addition of heparin.

To establish whether the heparin had similar effects in the context of other IgG formats, the Fab v2 was reformatted into IgG1, IgG2 and IgG4 isotypes. The authors determined that IgG1 and IgG4 were potent inhibitors, but IgG2 was not. To sort out what was happening, a panel of linker arm hinge mutants was constructed that were designed to alter hinge length and flexibility. This systematic search allowed the authors to hone-in on a sweet spot of hinge length and composition that maximized the effective inhibition of the antibody. From these data, the authors hypothesize that binding of the two Fab arms of v2 to the two diagonally dispose subunits of tryptase, when presented in an optimized IgG scaffold construction, effectively induces conformational strain that results in disassembly of the enzyme's tetrameric ternary structure.

Overall, this manuscript describes a high quality piece of research that should elicit strong interest from the antibody engineering community, as well as the general reader. Below are comments that the authors might consider.

- 1) The differences in the antibody binding effects to β -tryptase with and without the stabilization by heparin are a big piece of the story. How much does heparin actually stabilize the tetramer? A simple DSF experiment could clarify this. Once dissociated, can the monomers reassociate into a functional tetramer?

We thank the reviewer for their comment and raising these questions. There is a very large body of published work that has addressed both heparin stabilization

as well as monomer reassociation to tetramer (essentially all of the key references are included in the review by Hallgren and Pejler FEBS J 273:1871 (2006), which we have now included in the paper. Specific references include Schwartz and Bradford JBC 261:7273 (1986); Alter et al. Biochem. J. 248:821 (1987); Ren et al. J Immunol 160:4561 (1998); Hallgren et al J Mol Biol 345:129 (2005) among many others. Another key paper published in 2007 provides a detailed mechanistic assessment of catalytically active forms of tryptase the role of heparin in tetramer/monomer equilibria and enzymatic activity (Schechter et al. Biochemistry 46:9615 (2007). Overall, the data in these publications have shown that heparin significantly stabilizes the tetramer and that monomers can reassociate into a functional tetramer. As expected, the extent of tetramer stabilization and enzymatic activity is dependent upon pH, salt, heparin concentration and length, temperature and other factors. While we agree that a DSF experiment could provide some additional data on heparin stabilization of the tetramer, we feel the large body of published data provides sufficient support.

We have included the following in the introduction of the revised paper.

“A detailed mechanistic assessment of the role of heparin in tetramer stability, monomer/tetramer equilibria and enzymatic activity has been previously described (Schechter et al. Biochemistry 2007). An excellent review on tryptases by Hallgren and Pejler provides additional insights into these aspects (Hallgren et al. FEBS J 2006).”

- 2) A significant amount of work was done systematically modifying the linker regions to produce a series of IgG variants. It was surprising to me that the structural information that was available from the Fab v1- β -tryptase complex was not better utilized to guide and explain the linker engineering.

At line 454 begins a discussion about the length of linker arm (defined by the P217-C226 distance) of an IgG. Maybe I'm missing something, but the Fab v1 complex structure should provide much more insight into what is taking place in the case of IgG v2 than relying on IgG1 (PDB 1HZH). The Fab v1 has to have virtually the same epitope as v2 and thus, even taking into consideration the authors' suggestion that engagement angles might differ, the v1 structure should provide a good model for the orientation of the Fab. This, in turn, should allow a determination of the directionality and pretty much the point where the beginning part of the linker (P217) exits the heavy chain constant domain. The coordinates are not released yet and therefore it is not possible to determine how far the chain can be traced to P217; nevertheless, any approximations based on a relevant structure should provide more reliable information to build from than a generic IgG structure.

We thank the reviewer for this comment and pointing out the confusion. We did use our Fab-tryptase complex for our modeling and measurement of the P217-P217 distance between two Fabs. However, since the upper hinge was not part of our Fab, we used the corresponding segment from 1HZH. Since this segment

is a flexible peptide, the 1HZH segment is representative of that distance that the upper hinge can span in other IgGs, such as E104. We have updated the text in the first paragraph of our “Proposed model of inhibition by E104” to clarify this point.

- 3) Molecular pliers- This conceptual model of the MOA of the inhibitory effect of E104.v2 is built on the premise that coupled allosteric effects together with steric stress generated by this antibody “pry” the tetramer apart. There seems to be two parts here. Fab v1 dissociates the tetramer, heparin rescues it. It is probably possible to back out the energies associated with the Fab induce dissociation, but assigning the energy contributed by the “stress” induced by the pliers part is harder to get at experimentally. This is more a comment than criticism, but it would be interesting to work out the contribution of the Fab v1 alone to put the pliers’ contribution in better context.

We thank the reviewer for this comment. We agree it would be interesting to explore and measure the energetics of each contribution in future work. We have added the following sentence in the discussion on this topic.

“Future studies aimed at measuring the energetics of this physical strain would be both informative and interesting.”

- 4) Table 1 reports the KDs of the hinge sequence variants. The caption suggests that the affinities are to β -tryptase, which one might assume relates to the tetramer. However, the SPR data were collected on the monomer, so this should be clarified. The accuracy/precision of the SPR data also seem to be overinterpreted. There is common agreement among biophysicists that the accuracy of the numbers spit out from SPR sensograms fitting programs have to be taken with a grain of salt for ultra-high affinity interactions. To suggest that KD differences in the 0.5 nM range can be measured to an accuracy to the 2nd decimal place is pure fantasy. The supplementary material should include the kinetics and the sensograms for at least 5 of the best variants. The dissociation data were collected on 3 serial dilutions over 600 secs. It would take additional serial dilutions and require considerably longer collection times to generate KD accuracies that would be statistically relevant to the ones reported here. All that said, determining and reporting binding constants to that level of accuracy does not add much to the conclusions. The authors should use their judgement about how to report these data to ensure that the small differences in affinity between the variants are not over interpreted.

We thank the reviewer for these helpful comments. Our interpretation of the SPR data is that all of the hinge variants exhibit very similar binding affinities. We have updated the text for clarification. We also have inserted original sensorgrams for six representative antibodies as Supplementary Figure 3 as suggested by the reviewer. To address the reviewer’s concern regarding the accuracy/precision of these ultra-high affinities, we have reduced the number of significant figures for

the K_D values in Table 1. Finally, we have updated the results section and Table 1 caption to reflect that all SPR experiments were performed against the β -tryptase monomer.

Reviewer 2:

The manuscript entitled " Bivalent Antibody Pliers: Inhibition of β -tryptase by an allosteric mechanism 1 dependent on the IgG hinge" represent a comprehensive functional and structural study that aims at deciphering the mechanism of action of antibodies targeting β -tryptase. It is a nice example of combining high resolution techniques such X-ray and EM giving rather static picture of each individuals and low resolution approaches like HDX-MS answering dynamic properties of the complex.

There are major issues that must be addressed regarding HDX-MS data reporting and major revision is required.

- 1) HDX-MS experiments were performed using a 1:1 mixture of β I-tryptase:Fab (incubated at 25°C for 1 h) while SEC analyses were performed at 1:2 β I-tryptase:Fab (incubated for 15 min at 25°C). Why not using the same ratio for these two experiments? What is the quality control of the HDX-MS sample? Any SEC or native MS to probe that most of the sample correspond to β I-tryptase:Fab complex, without extensive free β I-tryptase or Fab that would avoid proper HDX data interpretation.

All hinge variants have similar binding affinities, roughly equivalent to 0.5 nM, which means that a 1:1 complex has less than 2% unbound under the conditions used for the HDX MS experiments. A molar excess of Fab in the SEC experiment was needed to show a peak for the unbound Fab, otherwise, only one peak (the complex) would be present.

Numerous orthogonal experiments (included) show that 1:1 ratio at the conditions of the experiment gives nearly 100% bound. The strongest evidence comes from SPR experiments, where selected individual traces are now included (Supplementary Figure 3).

- 2) It would have been interesting to have online SEC-native MS measurements for direct identification of SEC species, more informative than 1D SDS-PAGE gel.

We thank the reviewer for this suggestion. While we agree that online SEC-native MS may more directly identify each species, we believe the control samples and SDS-PAGE to determine each identity are sufficient. These results still provide unequivocal data to arrive at the key conclusions stated below.

- a) Fab.v2, but not Fab.v1, induces dissociation of the tryptase tetramer into monomers (Fig. 3a vs. 3b).
- b) This dissociation is due to allosteric effects at the small interface (Fig. 3c).

c) The addition of heparin can prevent Fab E104.v2-induced dissociation (Fig. 3h).

- 3) References related to HDX-MS experiments are rather old and more generally HDX-MS data do not fulfil current recommendation of the HDX-MS community published in Nat Methods. 2019; 16(7): 595–602. I would suggest authors to update their material and method section and supplementary data, among which
- temperature at which the H/D reaction was performed (see reply 1 below).
 - pertinent LC-MS settings (for example, the LC gradient and flow rates, reversed-phase columns used, MS ion source parameters etc.) should be reported (see reply 1 below).
 - duration of the digestion, flow rate and the temperature that the digestion was conducted under (see reply 1 below).
 - detail of the time points (see reply 1 below).
 - Please include the search parameters used for peptide identification (sequence database used for peptide search, MS1 and MS2 tolerance, min and max peptide length, fixed and variable modifications, cleavage sites and number of missed cleavages if applicable, score and FDR thresholds) (see replies 1, 2 below).
 - threshold for significant differences in HDX (a threshold value interpreted as representing a significant difference in HDX between examined protein states based on the quantitative measure of repeatability. It is also beneficial to consider applying objective statistical analysis to bolster authors conclusions (see replies 2, 3 below).
 - When reporting explicitly on the change of HDX in a peptide, it is usual to present peptide uptake plots, plotting each labeling time with the per-peptide standard deviation (see reply 4 below). It might also give better overview for the reader to show H/D plots for all peptides (see reply 4 below). Authors suggest some rearrangement of the structure during the complex formation and showing the entire H/D data will shed a light on the protein dynamic not only selected part (see replies 3, 4 below).
 - Please explicitly state the mapping methodology on the X-ray 3D structure and at which time points data are depicted, based on a quantitative and statistical argument applied to the entire dataset (see reply 5 below).
 - Precise if HDX data were corrected for back exchange or not (see reply 1 below).

We thank the reviewer for their thoughtful and thorough comments. To address the comments and suggestions, we have made a number of changes in the way we report the exchange data. Please refer to the answers below.

Numeric responses below address numeric points noted above.

1. Material and methods sections have significant changes to include all of the requested information.

2. As is now described in the main and supplementary text, experiments were collected in triplicate; significant differences were those where observed differences in deuterium uptake exceeded propagated measurement error for that sampling time. As a second filter against false positive ID's, we required that any true difference must be statistically different on more than one overlapping peptide.

3. Peptide protection factors were determined as described previously (Walters, 2017, Analytical Chemistry) for each APO (PF(T1,T2)) and HOLO dataset (PF(v1,v2)), where T1/T2 refer to trypsin alone in either experiment, and v1/v2 refer to the versions of E104 used in either experiment, respectively. Significant differences were those where $PF(v1,v2)/PF(T1,T2)$ exceeded an error threshold defined by propagating the uncertainty ($\delta PF(v1,v2)/PF(v1,v2)$) of each individual PF calculation (See the Supplementary Note for a more detailed description). This enabled statistical validation of our conclusions, and eliminated small differences in variables such as pH and temperature that might alter chemical exchange rates. An excel file (Supplementary Data) has been added to the supplemental information with all PFs computed for both experiments along with a comparative analysis.

We thank the reviewer for the suggestion to perform a more rigorous statistical analysis of our comparative data. This significance analysis eliminated a few of the significant peptides used originally to manually define the sites involved (see #5) due to error propagation during normalization (see Supplementary Note); however, in the end, this more rigorous analysis generally agreed with the more refined manual analysis. This statistical analysis, created to bolster our manual conclusions at your suggestion and built on previous work (described above), will undoubtedly be useful in the future. We thank the reviewer for the suggestion to create it.

4. Two new supplementary figures (Supplementary Figures 9 (E104.v1) and 10 (E104.v2) contain every peptide that was used in the experiments with bound trypsin and with Apo sample). The mean of triplicates (discussed in 3) was taken as the value of each data point as shown and the error bars show the range of all measurements.

5. Manually, overlapping peptides in the two new supplementary figures were compared the residues involved were estimated by comparing all of the peptides that reported on a particular group of residues (with significant differences as determined using criteria provided in 2) above. This process may reduce the number of residues suspected to be involved from that of a full peptide, to a smaller subset, but can never increase the number of participating residues beyond that which is contained in the peptide.

Reviewer 3:

The manuscript by Maun et al. is a beautiful piece of molecular analysis of antibodies directed against the human mast cell tryptase. The manuscript is close follow up of their recently published also equally brilliant article in Cell. The focus of the article is primarily the use of a large set of hinge variants of a humanized monoclonal that targets a region in the interphase between the four subunits of the human mast cell tryptase where they show that divalent binding in combination with distance constrains are of major importance for the inactivation of the tryptase by pushing the subunits apart. I have few if any comments on the actual work which is a beautiful piece of molecular design showing the importance of the hinge for the function of various human IgG isotypes.

My major concern is at another level and that the background information. The authors almost entirely neglect the presence of the other mast cell granule proteases that are stored together with the tryptase and their effects. I think they need to mention both the chymase, the mast cell specific carboxypeptidase CPA3 and cathepsin G and their role in mast cell function as well as potential side effects of an anti-tryptase antibody.

It has recently been shown that the human mast cell chymase and the tryptase may act together to control excessive TH2 mediated immunity by cleaving a highly selected set of cytokines and chemokines. They almost exclusively cleave five potent TH2 cytokines out of a very large panel of cytokines and chemokines, including three of the most well known TH2 cytokines, IL-33, IL-18 and TSLP and also two additional that recently have been shown to act as potent TH2 inducing cytokines namely IL-15 and IL-21.

Knock out of the mouse counterpart of the human mast cell chymase mMCP-4 has also been shown to enhance sensitization giving further support of this potential effect of these proteases.

I understand that Genentech is a commercial company pushed by investors to come up with new drugs but when publishing in respectable international journals it is important also to include potential side effects of their efforts. I therefore suggest the authors also to include information stating the presence of tryptase as one out several mast cell specific or related proteases and that these act together upon mast cell activation and that there are potential side effects with long term treatment with anti-tryptase and/or anti chymase antibodies potentially could affect TH1/TH2 balance and actually increase sensitization to earlier allergen and potentially also initiate novel sensitization. Their cleavage of a number of neuropeptides and other peptide hormones and various toxins (Chymase and CPA3) may also be factors that are of importance when looking at ways to inhibit their activity.

It is a beautiful piece of work that however needs to be put into a broader context. There is most likely a reason why both the chymase and the tryptase

have been maintained in all mammalian lineages for what it appears more than 200-250 million years of mammalian evolution. Chymases are present in all three extant mammalian lineages with very similar cleavage specificities. The so far only exceptions are rabbit and guinea pig where their chymases have become strict Leu-ases and not classical chymases. The picture of the tryptases is less complete but the genes are present in all mammalian lineages although any detailed analysis have been performed on marsupial and monotreme tryptases. The conservation of both enzymes gives strong indications for evolutionary important and therefore conserved functions. Long term inhibition of these components of our immune system may thereby involve certain amount of risk, which I think needs to be addressed to give a more correct view of potential side effects of anti tryptase antibody treatment.

The low level or absence of immediate side effects after injection of the anti tryptase antibodies make the analysis of their activity highly interesting. However, during such studies of their potential beneficial clinical effects, potential long term negative effects should also be kept in mind not to put patients involved in such clinical tests at risk. Questions that I think needs to be addressed in the article.

We thank reviewer 3 for their overall very positive comments on our manuscript. Reviewer 3 would like us to expand the context to include other mast cell proteases (especially chymase based on the comments), mast cell biology, and potential risks that might arise in clinical setting. We have introduced other mast cell proteases (e.g. chymase) and associated biology (including some new references) to address the points raised by the reviewer. However, we feel that a more extensive discussion would detract from the focus on tryptase and anti-tryptase and the novel mechanism of action dependent upon the hinge region. We are also fairly reluctant to speculate on any clinical outcomes – either positive or negative – and prefer to wait for clinical data to support any conclusions. We do add a sentence that does address the overall hopes and pitfalls going forward. We did not introduce chymase with respect to extant mammalian lineages as we preferred to stay focused and deemed this may distract the reader. Some of this discussion is found in the two new references we added.

We have inserted the following text in the discussion:

However, other proteases in mast cells such as chymase and carboxypeptidase A3 also play significant roles in mast cell biology and perhaps disease as well (Refs 1 and 3-5 in revised manuscript). Recent reports have shown that mast cell proteases (tryptase and/or chymase) can cleave a variety of cytokine and chemokine substrates, including several well-known TH2-promoting cytokines IL-33, IL-18, TSLP, IL-15 and IL-21 (Fu, et al. J Immunol; Fu, et al. Int J Mol Sci 2019). How any downstream signaling by altered levels of these cytokines might affect any possible therapeutic benefit from anti-tryptase awaits further studies in patients.

REVIEWERS' COMMENTS

Reviewer #1 (Remarks to the Author):

The authors have made the revisions in the manuscript that considered the comments and suggestions I made to the original submission. I think it is a high quality piece of work and support its publication.

A. Kossiakoff

Reviewer #2 (Remarks to the Author):

Authors answered to all my questions.

My only suggestion would be to include a PRIDE submission number in the paper so that the HDX dataset could be publicaly available.

The paper is now ready for publication from my side.